# Adversarial Combinatorial Semi-bandits with Graph Feedback

Yuxiao Wen [1]

## Abstract

In combinatorial semi-bandits, a learner repeatedly selects from a combinatorial decision set of arms, receives the realized sum of rewards, and observes the rewards of the individual selected arms as feedback. In this paper, we extend this framework to include *graph feedback*, where the learner observes the rewards of all neighboring arms of the selected arms in a feedback graph $G$. We establish that the optimal regret over a time horizon $T$ scales as $\widetilde{\Theta}(S\sqrt{T} + \sqrt{\alpha ST})$, where $S$ is the size of the combinatorial decisions and $\alpha$ is the independence number of $G$. This result interpolates between the known regrets $\widetilde{\Theta}(S\sqrt{T})$ under full information (i.e., $G$ is complete) and $\widetilde{\Theta}(\sqrt{KST})$ under the semi-bandit feedback (i.e., $G$ has only self-loops), where $K$ is the total number of arms. A key technical ingredient is to realize a convexified action using a random decision vector with negative correlations. We also show that online stochastic mirror descent (OSMD) that only realizes convexified actions in expectation is suboptimal. In addition, we describe the problem of *combinatorial semi-bandits with general capacity* and apply our results to derive an improved regret upper bound, which may be of independent interest.

## 1. Introduction

Combinatorial semi-bandits are a class of online learning problems that generalize the classical multi-armed bandits (Robbins, 1952) and have a wide range of applications including multi-platform online advertising (Avadhanula et al., 2021), online recommendations (Wang et al., 2017), webpage optimization (Liu & Li, 2021), and online shortest path (György et al., 2007). In these applications, instead of taking an individual action, a set of actions is chosen at each time

(Cesa-Bianchi & Lugosi, 2012; Audibert et al., 2014; Chen et al., 2013). Mathematically, over a time horizon of length $T$ and for a fixed combinatorial budget $S$, a learner repeatedly chooses a (potentially constrained) combination of $K$ individual arms within the budget, i.e. from the following decision set

$$\mathcal{A}_0 \subseteq \mathcal{A} \equiv \{v \in \{0,1\}^K : \|v\|_1 = S\},$$

and receives a linear payoff $\langle v, r^t \rangle$ where $r^t \in [0,1]^K$ denotes the reward associated to each arm at time $t$. After making the decision at time $t$, the learner observes $\{v_a r_a^t : a \in [K]\}$ as the *semi-bandit feedback* or the entire reward vector $r^t$ under *full information*. When $S = 1$, it reduces to the multi-armed bandits with either the bandit feedback or full information. For $S > 1$, the learner is allowed to select $S$ arms at each time and collect the cumulative reward.

Under the adversarial setting for bandits (Auer et al., 1995), no statistical assumption is made about the reward vectors $\{r^t\}_{t \in [T]}$. Instead, they are (potentially) generated by an adaptive adversary. The objective is to minimize the expected *regret* of the learner's algorithm $\pi$ compared to the best fixed decision in hindsight, defined as follows:

$$\mathbb{E}[\mathsf{R}(\pi)] = \mathbb{E}\left[\max_{v_* \in \mathcal{A}_0} \sum_{t=1}^{T} \langle v_* - v^t, r^t \rangle\right] \quad (1)$$

where $v^t \in \mathcal{A}_0$ is the decision chosen by $\pi$ at time $t$. The expectation is taken over any randomness in the learner's algorithm and over the rewards, since the reward $r^t$ is allowed to be generated adaptively and hence can be random. Note that while the adversary can generate the rewards $r^t$ adaptively, i.e. based on the learner's past decisions, the regret in (1) is measured against a fixed decision $v_*$ assuming the adversary would generate the same rewards.

While the semi-bandit feedback has been extensively studied, the current literature falls short of capturing additional information structures on the rewards of the individual arms, except for the full information case. As a motivating example, consider the multi-platform online advertising problem, where the arms represent the (discretized) bids. At each round and on each platform, the learner makes a bid and receives zero reward on losing the auction and her surplus on winning the auction. In many ads exchange platforms,

[1]Courant Institute of Mathematical Sciences, New York University, New York, USA. Correspondence to: Yuxiao Wen <yuxiaowen@nyu.edu>.

*Proceedings of the 42nd International Conference on Machine Learning*, Vancouver, Canada. PMLR 267, 2025. Copyright 2025 by the author(s).

the winning bid is always announced, and hence the learner can compute the counterfactual reward for any bids higher than her chosen bid (Han et al., 2024). This additional information is not taken into account in the semi-bandit feedback.

Another example is the online recommendation problem, where the website plans to present a combination of recommended items to the user. The semi-bandit feedback assumes that the user's behavior on the displayed items will reveal no information about the undisplayed items. However, this assumption often ignores the semantic relationship between the items. For instance, suppose two items $i$ and $j$ are both tissue packs with similar prices. If item $i$ is displayed and the user clicks on it, a click is likely to happen if item $j$ were to be displayed. On the other hand, if item $i$ is a football and item $j$ is a wheelchair, then a click on one probably means a no-click on the other. Information of this kind is beneficial for the website planner and yet overlooked in the semi-bandit feedback.

To capture this rich class of combinatorial semi-bandits with additional information, we consider a more general feedback structure described by a directed graph $G = ([K], E)$ among the $K$ arms. We assume $G$ is *strongly observable*, i.e. for every $a \in [K]$, either $(a, a) \in E$ or $(b, a) \in E$ for all $b \neq a$. After making the decision $v \in \mathcal{A}_0$ at each time, the learner now observes the rewards associated to all neighboring arms of the selected arms in $v$:

$$\left\{ v_i r_i^t : \exists a \in [K] \text{ such that } v_a = 1 \text{ and } (a, i) \in E \right\}.$$

This graph formulation allows us to leverage information that is unexploited in the semi-bandit feedback.

Note that when $G$ is complete, the feedback structure corresponds to having full information; when $G$ contains only the self-loops, it becomes the semi-bandit feedback. In the presence of a general $G$, the exploration-exploitation trade-off becomes more complicated, and the goals of this paper are (1) to fully exploit this additional structure in the regret minimization and (2) to understand the fundamental learning limit in this class of problems.

### 1.1. Related work

The optimal regret of the combinatorial semi-bandits has drawn a lot of attention and has been extensively studied in the bandit literature. With linear payoff, Koolen et al. (2010) shows that the Online Stochastic Mirror Descent (OSMD) algorithm achieves near-optimal regret $\widetilde{\Theta}(S\sqrt{T})$ under full information. In the case of the semi-bandit feedback, Audibert et al. (2014) shows that OSMD achieves near-optimal regret $\widetilde{\Theta}(\sqrt{KST})$ using an unbiased estimator $\tilde{r}_a^t = v_a^t r_a^t / \mathbb{E}_{v^t}[v_a^t]$, where $v^t$ is the random decision selected at time $t$ and the expectation denotes the probability

of choosing arm $a$.[1] The transition of the optimal regret's dependence from $\sqrt{KS}$ to $S$, as the feedback becomes richer, remains a curious and important open problem.

Another type of feedback is the bandit or full-bandit feedback, which assumes only the realized payoff $\langle v, r^t \rangle$ is revealed (rather than the rewards for individual arms). In this case, the minimax optimal regret is $\widetilde{\Theta}(\sqrt{KS^3T})$ (Audibert et al., 2014; Cohen et al., 2017; Ito et al., 2019). This additional $S$ factor, compared to the semi-bandit feedback, matches the difference in the observations: in this bandit feedback, the learner obtains a single observation at each time, while in the semi-bandit the learner gains $S$ observations. When the payoff function is nonlinear in $v$, Han et al. (2021) shows that the optimal regret scales with $K^d$ where $d$ roughly stands for the complexity of the payoff function. More variants of combinatorial semi-bandits include the knapsack constraint (Sankararaman & Slivkins, 2018), the fractional decisions (Wen et al., 2015), and the contextual counterpart (Zierahn et al., 2023).

In the multi-armed bandits, multiple attempts have been made to formulate and exploit the feedback structure as feedback graphs since Mannor & Shamir (2011). In particular, the optimal regret is shown to be $\widetilde{\Theta}(\sqrt{\alpha T})$ when $T \geq \alpha^3$ (Alon et al., 2015; Eldowa et al., 2024) and is a mixture of $T^{1/2}$ and $T^{2/3}$ terms when $T$ is small due to the exploration-exploitation trade-off (Kocák & Carpentier, 2023). When the graph is only weakly observable, i.e. every node $a \in [K]$ has nonzero in-degree, the optimal regret is $\widetilde{\Theta}(\delta^{1/3}T^{2/3})$ (Alon et al., 2015). Here $\alpha$ and $\delta$ are the independence and the domination number of the graph $G$ respectively, defined in Section 1.3.

Instead of a fixed graph $G$, Cohen et al. (2016) and Alon et al. (2017) study time-varying graphs $\{G_t\}$ and show that an upper bound $\widetilde{O}\left(\sqrt{\sum_{t=1}^{T} \alpha_t}\right)$ can be achieved. Additionally, a recent line of research (Balseiro et al., 2023; Han et al., 2024; Wen et al., 2024) introduces graph feedback to the tabular contextual bandits, in which case the optimal regret depends on a complicated graph quantity that interpolates between $\alpha$ and $K$ as the number of contexts changes.

### 1.2. Our results

In this paper, we present results on combinatorial semi-bandits with a strongly observable feedback graph $G$ and the full decision set $\mathcal{A}_0 = \mathcal{A}$, while results on general $\mathcal{A}_0$ are discussed in Section 5.1 and 5.2. Our results are summarized in Table 1, and the main contribution of this

---

[1] Audibert et al. (2014) only argues there exists a particular decision subset $\mathcal{A}_0$ under which the regret is $\Omega(\sqrt{KST})$. The lower bound for $\mathcal{A}$ is given by Lattimore et al. (2018).

*Table 1.* Minimax regret bounds up to polylogarithmic factors. Our results are in bold.

| | Semi-bandit ($\alpha = K$) | **General feedback graph** $G$ | Full information ($\alpha = 1$) |
|---|---|---|---|
| Regret | $\widetilde{\Theta}(\sqrt{KST})$ | $\widetilde{\mathbf{\Theta}}(\mathbf{S}\sqrt{\mathbf{T}} + \sqrt{\boldsymbol{\alpha}\mathbf{ST}})$ | $\widetilde{\Theta}(S\sqrt{T})$ |

paper is four-fold:

1. We introduce the formulation of a general feedback structure using feedback graphs in combinatorial semi-bandits.

2. On the full decision set $\mathcal{A}$, we establish a minimax regret lower bound $\Omega(S\sqrt{T} + \sqrt{\alpha ST})$ that correctly captures the regret dependence on the feedback structure and outlines the transition from $\widetilde{\Theta}(S\sqrt{T})$ to $\widetilde{\Theta}(\sqrt{KST})$ as the feedback gets richer.

3. We propose a policy OSMD-G (OSMD under graph feedback) that achieves near-optimal regret under general directed feedback graphs and adversarial rewards. Importantly, we identify that sampling with negative correlations is crucial in achieving the near-optimal regret, and that the original OSMD is provably suboptimal.

4. We formulate the problem of combinatorial semi-bandits with general capacity in Section 4 and provide an improved regret by applying our results under graph feedback. This formulation may be of independent interest.

When the feedback graphs $\{G_t\}_{t \in [T]}$ are allowed to be time-varying, we can also obtain a corresponding upper bound. The upper bounds are summarized in the following theorem.

**Theorem 1.1.** *Consider the full decision set $\mathcal{A}$. For $1 \leq S \leq K$ and any strongly observable directed graph $G = ([K], E)$, there exists an algorithm $\pi$ that achieves regret*

$$\mathbb{E}[\mathsf{R}(\pi)] = \widetilde{O}\Big(S\sqrt{T} + \sqrt{\alpha ST}\Big).$$

*When the feedback graphs $\{G_t\}_{t \in [T]}$ are time-varying, the same algorithm $\pi$ achieves*

$$\mathbb{E}[\mathsf{R}(\pi)] = \widetilde{O}\left(S\sqrt{T} + \sqrt{S \sum_{t=1}^{T} \alpha_t}\right)$$

*where $\alpha_t = \alpha(G_t)$ is the independence number of $G_t$.*

This algorithm $\pi$ is OSMD-G proposed in Section 3.1. In OSMD-G, the learner solves for an optimal convexified action $x \in \text{Conv}(\mathcal{A})$ via mirror descent at each time $t$, using the past observations, and then realizes it (in expectation) via

selecting a random decision vector $v^t$. In the extreme cases of full information and semi-bandit feedback, the optimal regret is achieved as long as $v^t$ realizes the convexified action $x$ in expectation (Audibert et al., 2014). However, this realization in expectation alone is provably suboptimal under graph feedback, as shown later in Theorem 3.4.

Under a general graph $G$, the regret analysis for a tight bound crucially requires this random decision vector to have negative correlations among the arms, i.e. $\mathsf{Cov}(v_i^t, v_j^t) \leq 0$ for $i \neq j$, in addition to the realization of $x$ in expectation. Consequently, the following technical lemma is helpful in our upper bound analysis:

**Lemma 1.2.** *Fix any $1 \leq S \leq K$ and $x \in \text{Conv}(\mathcal{A})$. There exists a probability distribution $p$ over $\mathcal{A}$ that satisfies:*

1. *(Mean) $\forall i \in [K]$, $\mathbb{E}_{v \sim p}[v_i] = x_i$.*

2. *(Negative correlations) $\forall i \neq j$, $\mathbb{E}_{v \sim p}[v_i v_j] \leq x_i x_j$, i.e. any pair of arms $(i, j)$ is negatively correlated.*

*In particular, there is an efficient scheme to sample from $p$.*

This lemma is a corollary of Theorem 1.1 in Chekuri et al. (2009), and the sampling scheme is the randomized swap rounding (Algorithm 2). The mean condition guarantees that the convexified action is realized in expectation. The negative correlations essentially allow us to control the variance of the observed rewards in OSMD-G, thereby decoupling the final regret into two terms. Intuitively, the negative correlations imply a more exploratory sampling scheme; a more detailed discussion is in Section 3.1.

To show that OSMD-G achieves near-optimal performance, we consider the following minimax regret:

$$\mathsf{R}^* = \inf_{\pi} \sup_{\{r^t\}} \mathbb{E}[\mathsf{R}(\pi)] \tag{2}$$

where the inf is taken over all possible algorithms and the sup is taken over all potentially adversarial reward sequences. The following lower bound holds:

**Theorem 1.3.** *Consider any decision subset $\mathcal{A}_0 \subseteq \mathcal{A}$ and strongly observable graph $G$. When $T \geq \max\{S, \alpha^3/S\}$ and $S \leq K/2$, it holds that*

$$\mathsf{R}^* = \Omega\Big(S\sqrt{T \log(K/S)} + \sqrt{\alpha ST}\Big).$$

Our lower bound construction in the proof is stochastic, as is standard in the literature, and thus stochastic combinatorial semi-bandits will not be easier.

### 1.3. Notations

For $n \in \mathbb{N}$, denote $[n] = \{1, 2, \ldots, n\}$. The convex hull of $\mathcal{A}$ is denoted by $\mathrm{Conv}(\mathcal{A})$, and the truncated convex hull is defined by

$$\mathrm{Conv}_\epsilon(\mathcal{A}) = \{x \in \mathrm{Conv}(\mathcal{A}) : x_i \geq \epsilon \text{ for all } i \in [K]\}.$$

We use the standard asymptotic notations $\Omega, O, \Theta$ to denote the asymptotic behaviors up to constants, and $\widetilde{\Omega}, \widetilde{O}, \widetilde{\Theta}$ up to polylogarithmic factors respectively. Our results will concern the following graph quantities:

$$\alpha = \max\{|I| : I \subseteq [K] \text{ is an independent subset in } G\},$$
$$\delta = \min\{|D| : D \subseteq [K] \text{ is a dominating subset in } G\}.$$

In a graph $G$, $I \subseteq [K]$ is an independent subset if for any $i, j \in I$, $(i, j) \notin E$; and $D \subseteq [K]$ is a dominating subset if for any $u \in [K]$, there exists $i \in D$ such that $(i, u) \in E$. For each node $a \in [K]$, denote its out-neighbors in $G$ by

$$N_{\mathrm{out}}(a) = \{i \in [K] : (a, i) \in E\}$$

and its in-neighbors by

$$N_{\mathrm{in}}(a) = \{i \in [K] : (i, a) \in E\}.$$

Then for a binary vector $v \in \mathcal{A}$ that represents an $S$-arm subset of $[K]$, we denote its out-neighbors in $G$ by the union $N_{\mathrm{out}}(v) = \bigcup_{v_a=1} N_{\mathrm{out}}(a)$. Let $D \subseteq \mathbb{R}^d$ be an open convex set, $\overline{D}$ be its closure, and $F : \overline{D} \to \mathbb{R}$ be a differentiable, strictly convex function. We denote the Bregman divergence defined by $F$ as

$$D_F(x, y) = F(x) - F(y) - \langle \nabla F(y), x - y \rangle.$$

## 2. Regret lower bound

In this section, we sketch the proof of the lower bound in Theorem 1.3 and defer the complete proof to Appendix A. The idea is to divide this learning problem into $S$ independent sub-problems and present the exploration-exploitation trade-off under a set of hard instances to arrive at the final minimax lower bound.

Under the complete graph $G$, Koolen et al. (2010) already gives a lower bound $\Omega(S\sqrt{T}\log(K/S))$ by reducing the full information combinatorial semi-bandits to the full information multi-armed bandits with rewards ranging in $[0, S]$. This reduction argument, however, does not lead to the other $\Omega(\sqrt{\alpha ST})$ part of the lower bound. It constructs a multi-armed bandit policy from any given combinatorial

semi-bandit policy and shows they share the same expected regret. Thus the lower bound of one translates to that of the other. As soon as the feedback structure is not full information, the observations and thus the behaviors of the two policies no longer align.

To prove the second part, note that $\Omega(\sqrt{\alpha ST})$ only manifests in the lower bound when $S < \alpha$. In this case, we partition an independent subset $I \subseteq [K]$ of size $\alpha$ into $S$ subsets $I_1, \ldots, I_S$ of equal size $\lfloor \frac{\alpha}{S} \rfloor$ and embeds an independent multi-armed bandit hard instance in each $I_m$ for $m \in [S]$. The other arms $J = [K] \backslash I$ may be more informative but will incur large regret. Thus a good learner cannot leverage arms in $J$ due to the exploration-exploitation trade-off.

The learner then needs to learn $S$ independent sub-problems with $ST$ total number of arm pulls. If the learner is 'balanced' in the sense that for each sub-problem $m \in [S]$,

$$T_m(T) = \sum_{t=1}^T \sum_{a \in I_m} \mathbb{1}[a \text{ is pulled}] \approx T,$$

then the existing multi-armed bandit lower bound implies that the regret incurred in each sub-problem is $\Omega(\sqrt{\alpha T/S})$, thereby a total regret $\Omega(\sqrt{\alpha ST})$. While in our case the learner may arbitrarily allocate the arm pulls over the $S$ sub-problems, it turns out to be sufficient to focus on the 'balanced' learners via a stopping time argument proposed in Lattimore et al. (2018). Intuitively, if a learner devotes pulls $T_m(T) \gg T$ for some $m$, then he/she *must* suffers regret $\Delta(T_m(T) - T)$ where $\Delta$ is the reward gap in the hard instance, which leads to suboptimal performance.

## 3. A near-optimal algorithm

This section is structured as follows: In Section 3.1, we present our OSMD-G algorithm and highlight the choice of reward estimators and the sampling scheme that allow us to deal with general feedback graphs. Then we show that OSMD-G indeed achieves near-optimal regret $\widetilde{O}(S\sqrt{T} + \sqrt{\alpha ST})$ in Section 3.2. Finally, we argue in Section 3.3 that if the requirement of negative correlations is removed, OSMD-G would be suboptimal.

### 3.1. Online stochastic mirror descent with graphs

The overall idea of OSMD-G (Algorithm 1) is to perform a gradient descent step at each time $t$, based on unbiased reward estimators, in a dual space defined by a mirror mapping $F$ that satisfies the following:

**Definition 3.1.** Given an open convex set $D \subseteq \mathbb{R}^d$, a mirror mapping $F : \overline{D} \to \mathbb{R}$ satisfies

- $F$ is strictly convex and differentiable on $D$;

- $\lim_{x \to \partial D} \|\nabla F(x)\| = +\infty$.

While OSMD-G works with any well-defined mirror mapping, we will prove the desired upper bound in Section 3.2 for OSMD-G with the negative entropy $F(x) = \sum_{i=1}^{K}(x_i \log(x_i) - x_i)$ defined on $D = \mathbb{R}_+^K$. For this choice of $F$, the dual space $D^* = \mathbb{R}^K$ and hence (5) is always valid. In fact, (5) admits the explicit form

$$w^{t+1} = x^t \exp(\eta \tilde{r}^t).$$

Recall at each time $t$, for a selected decision $v^t \in \mathcal{A}$, the learner observes graph feedback $\{v_i^t r_i^t : i \in N_{\text{out}}(v^t)\}$. Based on this, we define the reward estimator for each arm $a \in [K]$ at time $t$ in (4). As we invoke a sampling scheme to realize $x^t$ in expectation, i.e. $\mathbb{E}_{v^t \sim p^t}[v^t] = x^t$, our estimator in (4) is unbiased.

A crucial step in OSMD-G is to sample a decision $v^t$ at each time $t$ that satisfies both the mean condition $\mathbb{E}_{v^t \sim p^t}[v^t] = x^t$ and the negative correlation $\mathbb{E}_{v^t \sim p^t}[v_i^t v_j^t] \leq x_i^t x_j^t$. Thanks to Lemma 1.2, both conditions are guaranteed for *all* possible target $x^t \in \text{Conv}(\mathcal{A})$ when we invoke Algorithm 2 as our sampling subroutine.[2] The description and details of Algorithm 2 are deferred to Appendix B.

While seemingly intuitive given that $\|v^t\|_1 = S$, we emphasize that the negative correlations $\mathbb{E}_{v^t \sim p^t}[v_i^t v_j^t] \leq x_i^t x_j^t$ do not necessarily hold and can be non-trivial to achieve. Consider the case $S = 2$. When $x^t = \frac{2}{K}\mathbf{1}$ is the uniform vector, a uniform distribution over all pairs satisfies the correlation condition, seeming to suggest the choice of $p(i, j) \propto x_i^t x_j^t$. However, when $x^t = (1, 0.8, 0.2)$, the only such solution is to sample the combination $\{1, 2\}$ with probability 0.8 and $\{1, 3\}$ with probability 0.2, suggesting a zero probability for sampling $\{2, 3\}$. A general strategy must be able to generalize both scenarios. From the perspective of linear programming, the correlation condition adds $\binom{K}{2}$ constraints to the original $K$ constraints (from the mean condition) in finding $p^t$, making it much harder to find a feasible solution.

Now we give an intuitive argument for why such distribution $p$ exists under $\mathcal{A}$ and how the structure of the latter helps. When $S = 1$, any distributions possess negative correlations. Inductively, let us suppose such distributions exist for $1, 2, \ldots, S - 1$. Then for a fixed target $x \in \text{Conv}(\mathcal{A})$, we can always find an index $i \in [K]$ such that $\sum_{j=1}^{i-1} x_j + c x_i = 1$ and $\sum_{j=i+1}^{K} x_j + (1-c)x_i = S - 1$ for some $c \in [0, 1]$. Namely, the target of size $S$ is partitioned into two sub-targets with ranges $[1, i]$ and $[i, K]$, each with sizes 1 and $S - 1$, and with an overlap on index $i$. We can then assign $v_i = 0$ with probability $1 - x_i$, to the first half $[1, i]$ with probability $c x_i$, and to $[i, K]$ with probability $(1 - c)x_i$. To obtain a final size $S$ solution, we draw $v'$ supported on $[1, i-1]$ with size 0 or 1 and $v''$ on $[i + 1, K]$

---

[2]The use of Algorithm 2 is not essential as long as one can guarantee the negative correlations in Lemma 1.2.

**Algorithm 1** Online Stochastic Mirror Descent under Graph Feedback (OSMD-G)

**Input:** time horizon $T$, decision set $\mathcal{A}$, arms $[K]$, combinatorial budget $S$, feedback graph $G$, a truncation rate $\epsilon \in (0, 1)$, a learning rate $\eta > 0$, a mirror mapping $F$ defined on a closed convex set $\overline{\mathcal{D}} \supseteq \text{Conv}_\epsilon(\mathcal{A})$.

**Initialize:** $x^1 \leftarrow \arg\min_{x \in \text{Conv}_\epsilon(\mathcal{A})} F(x)$.

**for** $t = 1$ **to** $T$ **do**

    Generate a combinatorial decision $v^t$ by Algorithm 2 with target $x^t$.

    Observe the feedback $\{r_a^t : a \in N_{\text{out}}(v^t)\}$.

    Denote

$$\hat{r}_a^t = \frac{\sum_{i \in N_{\text{in}}(a)} \mathbb{1}[v_i^t = 1](1 - r_a^t)}{\sum_{i \in N_{\text{in}}(a)} x_i^t}. \quad (3)$$

    Build the reward estimator for each $a \in [K]$:

$$\tilde{r}_a^t = 1 - \hat{r}_a^t. \quad (4)$$

    **if** $S = 1$ **then**

        Denote $U_t = \{a \in [K] : \hat{r}_a^t \leq \frac{1}{(K-1)\epsilon}\}$.

        Set $\bar{r}^t = 1 + \sum_{a \in U_t} x_a^t \hat{r}_a^t$.

        Set $\tilde{r}_a^t \leftarrow \tilde{r}_a^t - \bar{r}^t$ for all $a \in [K]$.

    **end**

    Find $w^{t+1} \in \mathcal{D}$ such that

$$\nabla F(w^{t+1}) = \nabla F(x^t) + \eta \tilde{r}^t. \quad (5)$$

    Project $w^{t+1}$ to the truncated convex hull $\text{Conv}_\epsilon(\mathcal{A})$:

$$x^{t+1} \leftarrow \arg\min_{x \in \text{Conv}_\epsilon(\mathcal{A})} D_F(x, w^{t+1}). \quad (6)$$

**end**

---

with size $S - 1$ or $S - 2$, conditioned on the assignment of $v_i$. For any $j_1 \in [1, i-1]$, $j_2 \in [i+1, K]$, and $i$, any two of them are negatively correlated because, at a high level, the presence of one 'reduces' the size budget of the other. The negative correlations among the first half $[1, i-1]$ and $[i+1, K]$ are guaranteed by the induction hypothesis of the existence of such distributions for solutions with size less than $S$. Finally, the structure of $\mathcal{A}$ ensures that our pieced-together solution is valid, i.e. lies in $\mathcal{A}$.

### 3.2. Regret upper bound

In the following theorem, we show that OSMD-G achieves near-optimal regret for a strongly observable time-invariant feedback graph. The proof for time-varying feedback graphs $\{G_t\}_{t \in [T]}$ only takes a one-line change in (11). It is clear that Theorem 3.2 implies Theorem 1.1.

**Theorem 3.2.** *Let the mirror mapping be $F(x) =$*

$\sum_{i=1}^{K}(x_i \log x_i - x_i)$. *When the correlation condition for $p^t$ is satisfied, the expected regret of Algorithm 1 is upper bounded by*

$$\mathbb{E}[\mathsf{R}(\text{Alg 1})] \le \epsilon KT + \frac{S \log(K/S)}{\eta} + \eta(6S + 4\alpha \log(4KS/(\epsilon\alpha)))T.$$

*In particular, with truncation $\epsilon = \frac{1}{KT}$ and learning rate $\eta = \sqrt{\frac{5S \log(K/S)}{(6S + 4\alpha \log(4SK^2T/\alpha))T}} = \widetilde{O}\left(\sqrt{\frac{S}{(S+\alpha)T}}\right)$, it becomes*

$$\mathbb{E}[\mathsf{R}(\text{Alg 1})] \le 1 + \beta\left(S\sqrt{T} + \sqrt{\alpha ST}\right)$$

*for $\beta = \sqrt{24 \log(K/S) \log(4SK^2T/\alpha)} = \widetilde{O}(1)$.*

*Proof.* We present the proof for the case $S \ge 2$ here. The proof for $S = 1$ is similar and is deferred to Appendix C due to space limit. Now fix any $v \in \mathcal{A}$. Let

$$v_\epsilon = \underset{v' \in \text{Conv}_\epsilon(\mathcal{A})}{\arg\min} \|v - v'\|_1$$

which satisfies $(v - v_\epsilon)^\top r^t \le \|v - v_\epsilon\|_1 \le K\epsilon$ since $r^t \in [0,1]^K$. We can decompose the regret as

$$\mathbb{E}\left[\sum_{t=1}^{T}(v - v^t)^\top r^t\right] = \mathbb{E}\left[\sum_{t=1}^{T}(v - v_\epsilon)^\top r^t + (v_\epsilon - v^t)^\top r^t\right]$$

$$\le \epsilon TK + \mathbb{E}\left[\sum_{t=1}^{T}(v_\epsilon - v^t)^\top r^t\right] \quad (7)$$

Standard OSMD analysis applied to the truncated convex hull $\text{Conv}_\epsilon(\mathcal{A})$ further bounds the second term in (7) as follows (see e.g. Theorem 3 in Audibert et al. (2014)).

$$\mathbb{E}\left[\sum_{t=1}^{T}(v - v^t)^\top r^t\right]$$

$$\le \epsilon TK + \frac{S \log(K/S)}{\eta} + \eta \mathbb{E}\left[\sum_{t=1}^{T}\sum_{a=1}^{K} x_a^t (\tilde{r}_a^t)^2\right]. \quad (8)$$

To bound the last term, we first use the non-negativity of $\hat{r}_a^t$, defined in (3), to further decompose it:

$$\sum_{t=1}^{T}\sum_{a=1}^{K} x_a^t (\tilde{r}_a^t)^2 = \sum_{t=1}^{T}\sum_{a=1}^{K} x_a^t (1 - \hat{r}_a^t)^2$$

$$\le \sum_{t=1}^{T}\sum_{a=1}^{K} x_a^t \left(1 + (\hat{r}_a^t)^2\right) \le ST + \underbrace{\sum_{t=1}^{T}\sum_{a=1}^{K} x_a^t (\hat{r}_a^t)^2}_{(A)}.$$

Now we proceed to bound term (A). Recall that $G$ is strongly observable, and let $U = \{a \in [K] : (a,a) \notin E\}$ be the set of nodes with no self-loops. On the set $U$ we have

$$\mathbb{E}\left[\sum_{t=1}^{T}\sum_{a \in U} x_a^t (\hat{r}_a^t)^2\right]$$

$$= \sum_{t=1}^{T}\sum_{a \in U} \mathbb{E}\left[x_a^t \left(\frac{\sum_{i \in N_{\text{in}}(a)} \mathbb{1}[v_i^t = 1](1 - r_a^t)}{\sum_{i \in N_{\text{in}}(a)} x_i^t}\right)^2\right]$$

$$\overset{(a)}{\le} \sum_{t=1}^{T}\sum_{a \in U} \mathbb{E}\left[x_a^t \left(\frac{\sum_{i \ne a} \mathbb{1}[v_i^t = 1]}{\sum_{i \ne a} x_i^t}\right)^2\right]$$

$$\overset{(b)}{\le} \sum_{t=1}^{T}\sum_{a \in U} \mathbb{E}\left[x_a^t \left(\frac{S}{S - 1}\right)^2\right]$$

$$\le 4 \sum_{t=1}^{T}\sum_{a \in U} \mathbb{E}[x_a^t] \le 4ST. \quad (9)$$

Here (a) is due to $r_a^t \in [0,1]$ and that, if $a \in U$, then $(i,a) \in E$ for all $i \ne a$, and (b) uses $\sum_{i \ne a} x_i^t = S - x_a^t \ge S - 1$. On the other hand, by the choice of $v^t$ in Algorithm 1, the random variables $v_i^t$ are negatively correlated. Thus for each $a \in [K]$, we can upper bound the second moment of the following sum:

$$\mathbb{E}_{v^t \sim p^t}\left[\left(\sum_{i \in N_{\text{in}}(a)} v_i^t\right)^2\right]$$

$$= \left(\sum_{i \in N_{\text{in}}(a)} \mathbb{E}_{v^t \sim p^t}[v_i^t]\right)^2 + \text{Var}\left(\sum_{i \in N_{\text{in}}(a)} v_i^t\right)$$

$$= \left(\sum_{i \in N_{\text{in}}(a)} \mathbb{E}_{v^t \sim p^t}[v_i^t]\right)^2 + \sum_{i \in N_{\text{in}}(a)} \text{Var}(v_i^t)$$

$$+ \sum_{\substack{i,j \in N_{\text{in}}(a) \\ i \ne j}} \text{Cov}(v_i^t, v_j^t)$$

$$\le \left(\sum_{i \in N_{\text{in}}(a)} x_i^t\right)^2 + \sum_{i \in N_{\text{in}}(a)} x_i^t. \quad (10)$$

Then on the set $U^c \equiv [K] \backslash U$, we have

$$\mathbb{E}\left[\sum_{t=1}^{T}\sum_{a \notin U} x_a^t (\hat{r}_a^t)^2\right]$$

$$\le \sum_{t=1}^{T}\sum_{a \notin U} \mathbb{E}\left[x_a^t \left(\frac{\sum_{i \in N_{\text{in}}(a)} \mathbb{1}[v_i^t = 1]}{\sum_{i \in N_{\text{in}}(a)} x_i^t}\right)^2\right]$$

$$\overset{(10)}{\le} \sum_{t=1}^{T}\sum_{a \notin U} \mathbb{E}\left[x_a^t \left(1 + \frac{1}{\sum_{i \in N_{\text{in}}(a)} x_i^t}\right)\right]$$

$$\le \sum_{t=1}^{T}\sum_{a \notin U} \mathbb{E}\left[x_a^t \left(1 + \frac{1}{\sum_{i \notin U : i \in N_{\text{in}}(a)} x_i^t}\right)\right]$$

$$\overset{(c)}{\leq} T\left(S + 4\alpha \log\left(\frac{4KS}{\epsilon\alpha}\right)\right). \tag{11}$$

Here (c) uses $\sum_{a=1}^{K} x_a^t \leq S$, Lemma F.2 on the restricted subgraph $G|_{U^c}$, and the fact that $\alpha(G|_{U^c}) = \alpha(G) = \alpha$. Combining (9) and (11) yields

$$\mathbb{E}\left[\sum_{t=1}^{T}\sum_{a=1}^{K} x_a^t (\tilde{r}_a^t)^2\right] \leq 6TS + 4T\alpha \log\left(\frac{4KS}{\epsilon\alpha}\right). \tag{12}$$

Finally, combining (12) with (8), we end up with the desired upper bound

$$\mathbb{E}[\mathsf{R}(\text{Alg 1})] \leq \epsilon KT + \frac{S\log(K/S)}{\eta}$$
$$+ \eta(6S + 4\alpha \log(4KS/(\epsilon\alpha)))T.$$

$\square$

Note that at each time $t$ and for each arm $a \in [K]$, the total number of arms that observe $a$ is a random variable due to the random decision $v^t$. In (10) in the proof above, one can naively bound the second moment of this random variable by

$$\mathbb{E}\left[\left(\sum_{i\in N_{\text{in}}(a)} v_i^t\right)^2\right] \leq S\mathbb{E}\left[\sum_{i\in N_{\text{in}}(a)} v_i^t\right]$$

since $\|v^t\|_1 \leq S$, which leads to an upper bound $\widetilde{O}(S\sqrt{\alpha T})$. We will see that this rate is sometimes not improvable for certain proper decision subsets $\mathcal{A}_0 \subsetneq \mathcal{A}$ later in Section 5.1.

To improve on this bound for $\mathcal{A}$, we need to further exploit the structures of the full decision set $\mathcal{A}$ and the sampling distribution $p^t$ of $v^t$, which motivates Lemma 1.2. The negative correlations therein allow us to decompose this second moment into the squared mean and a sum of the individual variances, as in (10). By saving on the $O(K^2)$ correlation terms, this decomposition shaves the factor in (10) from $S\alpha$ to $S+\alpha$, yielding the desired result $\widetilde{O}(S\sqrt{T} + \sqrt{\alpha ST})$.

*Remark* 3.3. It turns out that when $S \geq 2$ and $G$ is strongly observable, the presence of the nodes with no self-loop can be easily handled in this upper bound analysis, whereas the case $S = 1$ proved in Appendix C requires more care. This matches the intuition that, when $S \geq 2$, the learner always observes the entire subset $U$ at every time $t$. Therefore, the extension from $U = \varnothing$ to $|U| \geq 1$ does not add to the difficulty in learning.

### 3.3. The necessity of negative correlations

The previous section shows an improved performance for OSMD-G when $v^t$ has negative correlations, which is a requirement never seen in either the semi-bandit feedback or the full feedback in previous literature. In either of the

two cases, OSMD with the mean condition (in Lemma 1.2) alone is sufficient to achieve the near-optimal regret.

Then, one may naturally ask if the vanilla OSMD-G with only the mean condition still achieves this improved rate, i.e. when it only guarantees $\mathbb{E}_{v^t \sim p^t}[v^t] = x^t$. The answer is negative.

**Theorem 3.4.** *Fix any problem parameters $(K, S, \alpha, T)$ with $S\alpha \leq K$, $S \leq \frac{K}{2}$, and $T \geq \max\{S, \alpha^3\}$, and consider the full decision set $\mathcal{A}$. There exists a feedback graph $G = ([K], E)$ and a sampling scheme $p^t$ that satisfies $\mathbb{E}_{v^t \sim p^t}[v^t] = x^t$, such that*

$$\sup_{\{r^t\}} \mathbb{E}[\mathsf{R}(\pi_0)] = \Omega\left(S\sqrt{\alpha T}\right)$$

*where $\pi_0$ denotes OSMD-G equipped with this $p^t$ and mirror mapping $F(x) = \sum_{i=1}^{K}(x_i \log x_i - x_i)$.*

*Proof.* The core idea of this proof is that, for some $G$ and $p^t$, running the vanilla OSMD-G on this problem instance is equivalent to running OSMD on a multi-armed bandit with rewards ranging in $[0, S]$. Without loss of generality, assume $K = nS$ for $n \in \mathbb{N}$.[3] By assumption $\alpha \leq n$.

First, we construct the graph $G$. Let $V_1, \ldots, V_n$ partition the nodes $[K]$ each with size $S$, and let $H = ([n], E_n)$ be an arbitrary graph on $n$ nodes with independence number $\alpha(H) = \alpha$. Then we let $(a, b) \in E$ iff either $a, b \in V_i$ or $a \in V_i, b \in V_j$, and $(V_i, V_j) \in E_n$, i.e. each $V_i$ is a clique and $H$ is a graph over the cliques.

For clarity, we denote the mean condition as

$$\mathbb{E}_{v^t \sim p^t}[v^t] = x^t \tag{M}$$

and for vector $q \in \mathbb{R}^K$, we say $q$ aligns with the cliques if

$$q_a = q_b \equiv q(V_i), \quad \forall a, b \in V_i \quad \forall i \in [n]. \tag{AC}$$

Now we consider a sampling scheme $p^t$ as follows: (1) if $x^t$ satisfies (AC), then let $v^t = V_i$ with probability $x^t(V_i)$; (2) otherwise, use any distribution $p^t$ satisfying (M). Note that (1) gives a valid distribution over the cliques and satisfies (M). We will show via an induction that if $r^t$ satisfies (AC) for all $t \in [T]$, then (2) never happens. As the base case, the OSMD initialization $x^1 = \frac{1}{K}\mathbf{1}$ satisfies (AC).

For the inductive step, when $x^t$ satisfies (AC), we have $v^t = V_i$ for some $i$ and thereby satisfies (AC). By construction of $G$, the reward estimator $\tilde{r}^t$ also satisfies (AC). Given the negative entropy mapping $F$, straightforward computation shows that both $w^{t+1}$ and $x^{t+1}$ satisfy (AC), completing the induction. Consequently, we have $v^t = V_{i_t}$ for some

---

[3]If $S$ does not divide $K$, one can put the remainder nodes in one of the cliques and slightly change the sampling $p^t$ to draw uniformly within this clique, while maintaining the mean condition.

$i_t \in [n]$ when $r^t$ satisfies (AC) for all $t \in [T]$. Namely, OSMD-G now reduces to a policy running on an $n$-armed bandit with feedback graph $H$, and now the lower bound of the latter can apply.

From the lower bound of the multi-armed bandits with feedback graphs (see e.g. Alon et al. (2015)), there exists a set of reward sequences $\{h^t(j)\}_{t \in [T], j \in \mathcal{J}}$ with some index set $\mathcal{J}$ and $h^t(j) \in [0, S]^n$ such that

$$\mathbb{E}_{j \sim \mathsf{Unif}(\mathcal{J})}[\mathsf{R}_{j,\mathsf{MAB}}(\pi)] = \Omega(S\sqrt{\alpha T})$$

for any policy $\pi$, where $\mathsf{R}_{j,\mathsf{MAB}}(\pi)$ denotes the multi-armed bandit regret when the reward sequence is $\{h^t(j)\}_{t, \in [T]}$. Define the clique-averaged reward sequences by $r_a^t(j) = \frac{h_i^t(j)}{|V_i|} \in [0, 1]$ for $a \in V_i$ for each $j \in \mathcal{J}$. Since (AC) is guaranteed, we have

$$\sup_{\{r^t\}} \mathbb{E}[\mathsf{R}(\pi_0)] \geq \mathbb{E}_{j \sim \mathsf{Unif}(\mathcal{J})}[\mathsf{R}_j(\pi_0)] = \Omega(S\sqrt{\alpha T})$$

where $\mathsf{R}_j(\pi_0)$ denotes the regret for this vanilla OSMD-G $\pi_0$ under reward sequence $\{r^t(j)\}_{t \in [T]}$. □

We remark that Theorem 3.4 does not directly show the necessity of negative correlations, even though they are sufficient as shown by Theorem 1.1. It only says that the mean condition alone is insufficient when dealing with general graph feedback, despite its success in the existing literature. It is possible that imposing extra conditions other than negative correlations can also lead to the near-optimal regret.

## 4. Solving semi-bandits with general capacity

In this section, we introduce a natural extension of combinatorial semi-bandits and show how we derive a near-optimal regret by applying the graph feedback. Specifically, consider the semi-bandit feedback where the learner observes $\{v_a r_a^t : a \in [K]\}$, but now each arm $a$ can be selected for at most $n_a \geq 1$ times, i.e. the decision set becomes

$$\mathcal{A} \equiv \{v \in [n_1] \times [n_2] \times \cdots \times [n_K] : \|v\|_1 = S\}.^4$$

Existing results do not directly apply to this extension. Instead, one can consider the equivalent problem with $N \equiv \sum_{a=1}^K n_a$ arms by having $n_a$ copies of arm $a$ for each $a \in [K]$, with the special structure that $r_{a(i)}^t = r_{a(j)}^t$ for all $t \in [T]$ and $i, j \in [n_a]$, when $\{a(i) : i \in [n_a]\}$ are the copies of $a$. If we simply take the upper bound for the semi-bandit feedback from Audibert et al. (2014) and ignore this structure, we arrive at regret $\widetilde{O}(\sqrt{NST}) = \widetilde{O}\left(\sqrt{ST\sum_{a=1}^K n_a}\right)$.

---

[4] As a motivation, consider dynamic allocations with $S$ units of resource at each time, and the $K$ arms have different capacities for the amount of resource they can consume and transform to utility. Their capacities can even be time-varying, by Theorem 1.1.

On the other hand, thanks to this special structure, there is a feedback graph that consists of $K$ cliques, each with $n_a$ nodes. Then we can exploit it using Algorithm 1 proposed in this work. Applying Theorem 1.1 leads to the regret $\widetilde{\Theta}(\sqrt{KST})$ (when $K \geq S$; otherwise it is $\widetilde{\Theta}(S\sqrt{T})$), which is near-optimal when $n_a = 1$ for $\Omega(K)$ arms following Theorem 1.3. Remarkably, although each arm has a different 'consumption' capacity, the regret characterization remains the same. This crucially relies on exploiting the feedback structure present in this equivalent formulation.

## 5. Extension to general decision subsets

### 5.1. When negative correlations are impossible

So far, we have shown the optimal regret $\widetilde{\Theta}(S\sqrt{T} + \sqrt{\alpha ST})$ on the full decision set $\mathcal{A}$. Our upper bound in Theorem 1.1 fails on general decision subsets $\mathcal{A}_0 \subseteq \mathcal{A}$, because it is not always possible to find a distribution $p^t$ for the decision $v^t$ in OSMD-G that provides the negative correlations in Lemma 1.2. For example, when there is a pair of arms $(a, b)$ with $v_a = v_b$ for all $v \in \mathcal{A}_0$, it is simply impossible to achieve negative correlations.

This failure, however, is not merely an analysis artifact. In the following, we present an example where moving from the full set $\mathcal{A}$ to a proper subset $\mathcal{A}_0 \subsetneq \mathcal{A}$ provably increases the optimal regret to $\widetilde{\Theta}(\min\{S\sqrt{\alpha T}, \sqrt{KST}\})$ when $S \leq \frac{K}{2}$. This argument is very similar to the proof of Theorem 3.4.

We first consider the case $S\alpha \leq K$. Assume again $S \leq \frac{K}{2}$ and $S$ divides $K$. We let $V_1, V_2, \ldots, V_{K/S}$ be a partition of the arms $[K]$ of equal size $S$. For the feedback graph $G$, let each $V_i$ be a clique for $i = 1, \ldots, K/S$. Let $H = (\{V_1, \ldots, V_{K/S}\}, \overline{E})$ be an arbitrary other graph over the cliques such that $(V_i, V_j) \in \overline{E}$ in $H$ iff $(a, b) \in E$ for all $a \in V_i$ and $b \in V_j$ in $G$. The independence numbers $\alpha(G) = \alpha(H)$ are equal. On the full decision set $\mathcal{A}$, Theorem 1.1 and 1.3 tell us the optimal regret is $\widetilde{\Theta}(S\sqrt{T} + \sqrt{\alpha ST})$.

Now consider a proper decision subset

$$\mathcal{A}_{\mathrm{partition}} = \{\mathbf{1}_{1:S}, \mathbf{1}_{S+1:2S}, \ldots, \mathbf{1}_{K-S+1:K}\} \quad (13)$$

where $(\mathbf{1}_{i:j})_k = \mathbb{1}[i \leq k \leq j]$ is one on the coordinates from $i$ to $j$ and zero otherwise. Namely, the only feasible decisions are the first $S$ arms in $V_1$, the next $S$ arms in $V_2$, ..., and the last $S$ arms in $V_{K/S}$. It is straightforward to see that this problem is equivalent to a multi-armed bandit with $K/S$ arms and a feedback graph $H$, and the rewards range in $[0, S]$. From the bandit literature (Alon et al., 2015), the optimal regret on this decision subset $\mathcal{A}_{\mathrm{partition}}$ is $\widetilde{\Theta}(S\sqrt{\alpha T})$ which is fundamentally different from the result for the full decision set, even under the same feedback graph.

On the other hand, if $S\alpha > K$, a similar construction fol-

lows, except that some of the grouped nodes $V_i$ are no longer cliques in order to satisfy $\alpha(G) = \alpha$, and that the graph $H$ has only self-loops. Then $\alpha(H) = \frac{K}{S}$ and the regret is $\widetilde{\Theta}(\sqrt{KST})$. To formalize this statement:

**Theorem 5.1.** *Fix any problem parameters $(K, S, \alpha, T)$ with $S\alpha \leq K$, $S \leq \frac{K}{2}$, and $T \geq \max\{S, \alpha^3\}$. There exists a decision subset $\mathcal{A}_0 \subsetneq \mathcal{A}$ such that*

$$\mathsf{R}^*(\mathcal{A}_0) = \Omega\Big(\min\{S\sqrt{\alpha T}, \sqrt{KST}\}\Big)$$

*where $\mathsf{R}^*(\mathcal{A}_0)$ denotes the minimax regret, as defined in (2), on this subset $\mathcal{A}_0$.*

Given this (counter-)example, the following upper bound is of interest:

**Theorem 5.2.** *On general decision subset $\mathcal{A}_0 \subseteq \mathcal{A}$ where only the mean condition is guaranteed, the algorithm OSMD-G achieves*

$$\mathbb{E}[\mathsf{R}(\mathrm{Alg}\ 1)] = \widetilde{O}\Big(S\sqrt{\alpha T}\Big).$$

*In particular, when $S\alpha > K$, one can ignore the graph feedback and directly apply OSMD. The combination of OSMD and OSMD-G then guarantees $\widetilde{O}\Big(\min\{S\sqrt{\alpha T}, \sqrt{KST}\}\Big)$.*

For any target $x^t \in \mathrm{Conv}(\mathcal{A}_0)$, there is always a probability distribution $p^t$ such that $\mathbb{E}_{v^t \sim p^t}[v^t] = x^t$, which is used in earlier works (Koolen et al., 2010; Audibert et al., 2014). With this choice of $p^t$, OSMD-G achieves the regret in Theorem 5.2. The proof follows from Section 3.2 and is left to Appendix E. Together with the construction of $\mathcal{A}_{\mathrm{partition}}$ in (13), it suggests that leveraging the negative correlations, whenever the decision subset $\mathcal{A}_0$ allows, is crucial to achieving improved regret $\widetilde{O}(S\sqrt{T} + \sqrt{\alpha ST})$. We will see examples of $\mathcal{A}_0$ where negative correlations are guaranteed in the next section.

Note on general $\mathcal{A}_0$, the efficiency of OSMD-G is no longer guaranteed; see discussions in Koolen et al. (2010); Audibert et al. (2014). To compensate, we provide an efficient elimination-based algorithm that is agnostic of the structure of the decision subset $\mathcal{A}_0$ and achieves $\widetilde{O}(S\sqrt{\alpha T})$ when the rewards are stochastic. The algorithm and its analysis are left in Appendix D.

### 5.2. When negative correlations are possible

This section aims to extend the upper bound in Theorem 1.1 to some other decision subsets $\mathcal{A}_0 \subseteq \mathcal{A}$. First, by Theorem 1.1 in Chekuri et al. (2009), Lemma 1.2 and OSMD-G can be generalized directly to any decision subset $\mathcal{A}_0' \subseteq \{v \in \{0,1\}^K : \|v\|_1 \leq S\}$ that forms a matroid. Notably, matroids require that decisions with size less than $S$ are also feasible, hence they are different from the setup $\mathcal{A}_0 \subseteq \mathcal{A}$ we consider throughout this work.

In addition, while Chekuri et al. (2009) focuses on matroids, the proof of their Theorem 1.1 only relies on the following *exchange property* of a decision set $\mathcal{A}_0$: for any $v, u \in \mathcal{A}_0$, there exist $i \in u - v$ and $j \in v - u$ such that $u - \{i\} + \{j\}, v - \{j\} + \{i\} \in \mathcal{A}_0$. Lemma 1.2 remains valid for any such $\mathcal{A}_0$. Here we provide an example of $\mathcal{A}_0 \subsetneq \mathcal{A}$ that satisfies this property:

Consider the problem that the learner operates on $S$ systems in parallel, and on each system $s$ he/she has $K_s$ arms to choose from. Then $K = \sum_{s \in [S]} K_s$ and the feasible decisions are $\mathcal{A}_0 = \{(v_1, \ldots, v_S) : v_s \in [K_s]\}$. It is clear that this $\mathcal{A}_0$ satisfies the exchange property above, and hence OSMD-G and Theorem 1.1 apply directly to such problems. The independence number $\alpha$ can be small if there is shared information among the $S$ systems.

### 5.3. Other open problems

**Weakly observable graphs:** The results in this work focus on the strongly observable feedback graphs. A natural extension would be the minimax regret characterization when the feedback graph $G = ([K], E)$ is only weakly observable. Recall that when $S = 1$, Alon et al. (2015) shows the optimal regret is $\widetilde{\Theta}(\delta^{1/3}T^{2/3})$.

To get a taste of it, consider a simple explore-then-commit (ETC) policy under stochastic rewards: the learner first explores the arms in a minimal dominating subset as uniformly as possible for $T_0$ time steps, and then commit to the $S$ empirically best arms for the rest of the time.[5] Its performance is characterized by the following result.

**Theorem 5.3.** *With high probability, the ETC policy achieves regret $\widetilde{O}(ST^{2/3} + \delta^{1/3}S^{2/3}T^{2/3})$.*

When $S = 1$, this policy is near-optimal. We briefly outline the proof here. When $\delta \geq S$, thanks to the stochastic assumption and concentration inequalities, each one of the $S$ empirically best arms contributes only a sub-optimality of order $\widetilde{O}(\sqrt{\delta/ST_0})$ with high probability. Trading off $T_0$ in the upper bound $ST_0 + ST\sqrt{\delta/(ST_0)}$ gives the bound $\widetilde{O}(\delta^{1/3}S^{2/3}T^{2/3})$. When $\delta < S$, a similar analysis yields the bound $\widetilde{O}(ST^{2/3})$.

**Problem-dependent bounds:** With the semi-bandit feedback and stochastic rewards, Combes et al. (2015) proves a problem-dependent bound $\widetilde{O}\Big(\frac{K\sqrt{S}}{\Delta_{\min}}\Big)$ where $\Delta_{\min}$ denotes the mean reward gap between the best decision and the second-best decision, or equivalently the $S$-th best arm and the $(S+1)$-th under the full decision set $\mathcal{A}$. It would be another interesting question to see how the presence of feedback graph $G$ helps the problem-dependent bounds.

---

[5]While finding the minimal dominating subset is NP-hard, there is an efficient $\log(K)$-approximate algorithm, which we include in Appendix F for completeness.

# Acknowledgements

The author is grateful to Yanjun Han for very helpful discussions and pointing to the matroid literature. The author also thanks anonymous reviewers for pointing out a flaw in an earlier version of the lower bound proof and for the advice to present an impossibility result for OSMD without negative correlations.

# Impact Statement

This paper presents work whose goal is to advance the field of Machine Learning and the theoretical understanding of Online Learning. There are many potential societal consequences of our work, none of which we feel must be specifically highlighted here.

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

## A. Proof of Theorem 1.3

Under the full information setup (i.e. when $G$ is a complete graph), a lower bound $\Omega(S\sqrt{T\log(K/S)})$ was given by (Koolen et al., 2010), which implies that $\mathsf{R}^*(G) = \Omega(S\sqrt{T\log(K/S)})$ for any general graph $G$. Note the assumption $S \le K/2$ is used in their proof to reduce the $K$ arms into an instance of multi-armed bandits with full information and $\lfloor\frac{K}{S}\rfloor$ arms, which then gives the desired lower bound.

To show the second part of the lower bound, without loss of generality, we may assume $\alpha = nS$ for some $n \in \mathbb{N}_{\ge 4}$. Consider a maximal independent set $I \subseteq [K]$ and partition it into $I_1, \ldots, I_S$ such that $|I_m| = n = \frac{\alpha}{S}$ for $m \in [S]$. Index each subset by $I_m = \{a_{m,1}, \ldots, a_{m,n}\}$. To construct a hard instance, let $u \in [n]^S$ be a parameter and the product reward distribution be $P^u = \prod_{a\in[K]} \mathsf{Bern}(\mu_a)$ where

$$\mu_a = \begin{cases} \frac{1}{4} + \Delta & \text{if } a = a_{m,u_m} \in I_m \text{ for } m \in [S]; \\ \frac{1}{4} & \text{if } a \in I \backslash \{a_{m,u_m}\}_{m\in[S]}; \\ 0 & \text{if } a \notin I. \end{cases}$$

The reward gap $\Delta \in (0, 1/4)$ will be specified later. Also let $P^{u_{-m}}$ differ from $P^u$ at $\mu_a = \frac{1}{4}$ for all $a \in I_m$, where $u_{-m} = (u_1, \ldots, u_{m-1}, 0, u_{m+1}, \ldots, u_S)$ denotes the parameter $u$ with $m$-th entry replaced by $0$. Then the following observations hold:

1. For each $u \in [n]^S$, the optimal combinatorial decision is $v_*(u) = \{a_{m,u_m}\}_{m\in[S]}$, and any other $v \in \mathcal{A}$ suffers an instantaneous regret at least $\Delta|v\backslash v_*(u)|$;

2. For each $u$ or $u_{-m}$, a decision $v \in \mathcal{A}$ suffers an instantaneous regret at least $\frac{1}{4}|v \cap I^c|$;

Fix any policy $\pi$ and denote by $v_t$ the arms pulled by $\pi$ at time $t$. Let $N_{m,j}(t)$ be the number of times $a_{m,j}$ is pulled at the end of time $t$ and $N_m(t) = \sum_{j=1}^{n} N_{m,j}(t)$, and $N_0(t)$ be the total number of pulls outside $I$ at the end of time $t$. Let $u$ be uniformly distributed over $[n]^S$, $\mathbb{E}^{(u)}[\cdot]$ denote the expectation under environment $P^u$, and $\mathbb{E}_u[\cdot]$ denote the expectation over $u \sim \mathsf{Unif}([n]^S)$.

Define the stopping time by $\tau_m = \min\{T, \min\{t : T_m(t) \ge T\}\}$. Note that $T \le N_m(\tau_m) \le T + S$ since at each round the learner can pull at most $S$ arms in $I_m$. Under any $u$, the regret is lower bounded by:

$$\mathbb{E}^{(u)}[\mathsf{R}(\pi)] \ge \Delta\mathbb{E}^{(u)}\left[N_0(T) + \sum_{m=1}^{S} N_m(T) - N_{m,u_m}(T)\right] = \Delta\mathbb{E}^{(u)}\left[\sum_{m=1}^{S} T - N_{m,u_m}(T)\right]$$

$$\mathbb{E}^{(u)}[\mathsf{R}(\pi)] \ge \Delta\mathbb{E}^{(u)}\left[\sum_{m=1}^{S}\sum_{t=1}^{T}\sum_{j=1}^{n} \mathbb{1}[a_{m,j} \in v_t]\right] \ge \Delta\mathbb{E}^{(u)}\left[\sum_{m=1}^{S}\sum_{t=1}^{\tau_m}\sum_{j=1}^{n} \mathbb{1}[a_{m,j} \in v_t]\right]$$

$$= \Delta\mathbb{E}^{(u)}\left[\sum_{m=1}^{S} N_m(\tau_m) - N_{m,u_m}(\tau_m)\right].$$

Together with $x + y \ge \max\{x, y\}$, we have

$$\mathbb{E}^{(u)}[\mathsf{R}(\pi)] \ge \frac{\Delta}{2}\sum_{m=1}^{S}\mathbb{E}^{(u)}[\max\{T - N_{m,u_m}(T), N_m(\tau_m) - N_{m,u_m}(\tau_m)\}]$$

$$\ge \frac{\Delta}{2}\sum_{m=1}^{S}\mathbb{E}^{(u)}[T - N_{m,u_m}(\tau_m)]$$

where the second line follows from the definition of $\tau_m$. Next, we lower bound the worst-case regret by the Bayes regret:

$$\max_{u \in [n]^S} \mathbb{E}^{(u)}[\mathsf{R}(\pi)] \geq \mathbb{E}_u \mathbb{E}^{(u)}[\mathsf{R}(\pi)] \geq \frac{\Delta}{2} \sum_{m=1}^{S} \mathbb{E}_u \mathbb{E}^{(u)}[T - N_{m,u_m}(\tau_m)]$$

$$= \frac{\Delta}{2} \sum_{m=1}^{S} \mathbb{E}_{u_{-m}} \left[ \frac{1}{n} \sum_{u_m=1}^{n} \mathbb{E}^{(u)}[T - N_{m,u_m}(\tau_m)] \right]$$

$$= \frac{\Delta}{2} \sum_{m=1}^{S} \mathbb{E}_{u_{-m}} \left[ T - \frac{1}{n} \sum_{u_m=1}^{n} \mathbb{E}^{(u)}[N_{m,u_m}(\tau_m)] \right] \tag{14}$$

For any fixed $m$, $u_{-m}$, and $u_m \in [n]$, let $\mathbb{P}_m$ denote the law of $N_{m,u_m}(\tau_m)$ under environment $u$, and $\mathbb{P}_{-m}$ denote the law of $N_{m,u_m}(\tau_m)$ under environment $u_{-m}$. Then

$$\mathbb{E}^{(u)}[N_{m,u_m}(\tau_m)] - \mathbb{E}^{(u_{-m})}[N_{m,u_m}(\tau_m)] \overset{(a)}{\leq} T\sqrt{\frac{1}{2}\mathsf{KL}(\mathbb{P}_{-m} \| \mathbb{P}_m)}$$

$$\overset{(b)}{\leq} T\sqrt{\frac{32\Delta^2}{3}\mathbb{E}^{(u_{-m})}[N_0(\tau_m) + N_{m,u_m}(\tau_m)]}$$

$$\leq 4\Delta T\sqrt{\mathbb{E}^{(u_{-m})}[N_0(T)] + \mathbb{E}^{(u_{-m})}[N_{m,u_m}(\tau_m)]}.$$

Here (a) uses Pinsker's inequality, and (b) uses the chain rule of the KL divergence, the inequality $\mathsf{KL}(\mathsf{Bern}(p)\|\mathsf{Bern}(q)) \leq \frac{(p-q)^2}{q(1-q)}$ and $\Delta \in (0, 1/4)$, and the important fact that $T_{m,u_m}(\tau_m)$ is $\mathcal{F}_{\tau_m}$-measurable. The last fact crucially allows us to look at the KL divergence only up to time $\tau_m$.

Note that $\mathbb{E}^{(u_{-m})}[\mathsf{R}(\pi)] \geq \frac{1}{4}\mathbb{E}^{(u_{-m})}[N_0(T)]$. So if $\mathbb{E}^{(u_{-m})}[N_0(T)] \geq \sqrt{\alpha ST}$ for any $m \in [S]$, the policy incurs too large regret under this environment $u_{-m}$ and we are done. Now suppose $\mathbb{E}^{(u_{-m})}[N_0(T)] < \sqrt{\alpha ST}$ for every $m$. By Cauchy-Schwartz inequality and the definition of $\tau_m$,

$$\sum_{u_m=1}^{n} \mathbb{E}^{(u)}[N_{m,u_m}(\tau_m)] \leq \sum_{u_m=1}^{n} \mathbb{E}^{(u_{-m})}[N_{m,u_m}(\tau_m)] + 4\Delta T\sqrt{n^2\sqrt{\alpha ST} + n\sum_{u_m=1}^{n}\mathbb{E}^{(u_{-m})}[N_{m,u_m}(\tau_m)]}$$

$$\leq T + S + 4\Delta T\sqrt{n^2\sqrt{\alpha ST} + n(T+S)}. \tag{15}$$

Plugging (15) into (14) leads to

$$\max_{u \in [n]^S} \mathbb{E}^{(u)}[\mathsf{R}(\pi)] \geq \frac{\Delta}{2} \sum_{m=1}^{S} \mathbb{E}_{u_{-m}} \left[ T - \frac{T+S}{n} - 4\Delta T\sqrt{\sqrt{\alpha ST} + \frac{T+S}{n}} \right]$$

$$= \frac{\Delta ST}{2} - \frac{\Delta S(T+S)}{2n} - 2\Delta^2 ST\sqrt{\sqrt{\alpha ST} + \frac{T+S}{n}}$$

$$\overset{(c)}{\geq} \frac{\Delta ST}{4} - 4\Delta^2 ST\sqrt{\frac{T}{n}}.$$

where (c) uses the assumptions that $T \geq S$, $n \geq 4$, and $\frac{2T}{n} \geq \sqrt{\alpha ST}$ when $T \geq \frac{\alpha^3}{S}$. Plugging in $\Delta = \frac{1}{64}\sqrt{\frac{n}{T}}$ and recalling $n = \frac{\alpha}{S}$ yield the desired bound

$$\max_{u \in [n]^S} \mathbb{E}^{(u)}[\mathsf{R}(\pi)] \geq \frac{1}{1024}\sqrt{\alpha ST}.$$

Note that the constants in this proof are not optimized.

## B. Randomized Swap Rounding

This section introduces the randomized swap rounding scheme by Chekuri et al. (2009) that is invoked in Algorithm 1. Note that randomized swap rounding is not always valid for any decision set $A$: its validity crucially relies on the exchange

property that for any $u, c \in A$, there exist $a \in u \backslash c$ and $a' \in c \backslash u$ such that $u - \{a\} + \{a'\} \in A$ and $c - \{a'\} + \{a\} \in A$. This property is satisfied by the full decision set $\mathcal{A}$ as well as any subset $A \subseteq \{v \in \{0,1\}^K : \|v\|_1 \leq S\}$ that forms a matroid. However, for general $A$ this can be violated, and as discussed in Section 5.1, no sampling scheme can guarantee the negative correlations and the learner must suffer a $\widetilde{\Theta}(S\sqrt{\alpha T})$ regret.

---

**Algorithm 2** Randomized Swap Rounding

---

**Input:** decision set $A$, arms $[K]$, target $x = \sum_{i=1}^N w_i v_i$ where $N = |A|$.
**Initialize:** $u \leftarrow v_1$.
**for** $i = 1$ **to** $N - 1$ **do**
    Denote $c \leftarrow v_{i+1}$ and $\beta_i \leftarrow \sum_{j=1}^i w_j$.
    **while** $u \neq c$ **do**
        Pick $a \in u \backslash c$ and $a' \in c \backslash u$ such that $u - \{a\} + \{a'\} \in A$ and $c - \{a'\} + \{a\} \in A$.
        With probability $\frac{\beta_i}{\beta_i + w_{i+1}}$, set $c \leftarrow c - \{a'\} + \{a\}$;
        Otherwise, set $u \leftarrow u - \{a\} + \{a'\}$.
    **end**
**end**
Output $u$.

---

# C. Case $S = 1$ in the proof of Theorem 3.2

In this section, we present the proof of Theorem 3.2 for the special case $S = 1$. The overall idea is the same as in Section 3.2 but requires an adaptation of Lemma 4 in Alon et al. (2015) to our reward setting.

*Proof.* Let $U = \{a \in [K] : (a, a) \notin E\}$. For the clarity of notation, let $\tilde{r}_a^t$ be defined as in (4) and recall $\bar{r}^t = 1 + \sum_{a \in U} x_a^t \hat{r}_a^t \geq 0$. Fix any $v \in \mathcal{A}$ and let $v_\epsilon = \arg\min_{v' \in \text{Conv}_\epsilon(\mathcal{A})} \|v - v'\|_1$. The regret becomes

$$\mathbb{E}\left[\sum_{t=1}^T (v - v^t)^\top r^t\right] \leq \epsilon KT + \mathbb{E}\left[\sum_{t=1}^T (v_\epsilon - v^t)^\top r^t\right] = \epsilon KT + \mathbb{E}\left[\sum_{t=1}^T (v_\epsilon - v^t)^\top (r^t - c_t \mathbf{1})\right]$$

for any $c_t \in \mathbb{R}$ when $S = 1$, where $\mathbf{1} \in \mathbb{R}^K$ denotes the all-one vector. Recall that $\tilde{r}_a^t$ is an unbiased estimator of $r_a^t$ and plug in $c_t = \bar{r}^t$, we get

$$\mathbb{E}\left[\sum_{t=1}^T (v - v^t)^\top r^t\right] \leq \epsilon KT + \mathbb{E}\left[\sum_{t=1}^T (v_\epsilon - v^t)^\top (\tilde{r}^t - \bar{r}^t \mathbf{1})\right].$$

Following the same lines in the proof of Theorem 3.2, we arrive at a similar decomposition as (8):

$$\mathbb{E}\left[\sum_{t=1}^T (v_\epsilon - v^t)^\top (\tilde{r}^t - \bar{r}^t \mathbf{1})\right] \leq \frac{S \log(K/S)}{\eta} + \eta \mathbb{E}\left[\sum_{t=1}^T \sum_{a=1}^K x_a^t (\tilde{r}_a^t - \bar{r}^t)^2\right]. \tag{16}$$

Now for any time $t$, it holds that

$$\sum_{t=1}^T \sum_{a \in U_t} x_a^t (\tilde{r}_a^t - \bar{r}^t)^2 = \sum_{t=1}^T \sum_{a \in U_t} x_a^t (\hat{r}_a^t + \bar{r}^t - 1)^2$$

$$= \sum_{t=1}^T \sum_{a \in U_t} x_a^t (\hat{r}_a^t)^2 - \sum_{t=1}^T \left(\sum_{a \in U_t} x_a^t \hat{r}_a^t\right)^2$$

$$\leq \sum_{t=1}^T \sum_{a \in U} x_a^t (\hat{r}_a^t)^2 - \sum_{t=1}^T \sum_{a \in U} (x_a^t)^2 (\hat{r}_a^t)^2$$

$$= \sum_{t=1}^T \sum_{a \in U} x_a^t (1 - x_a^t) (\hat{r}_a^t)^2$$

where the inequality is due to the non-negativity of $x_a^t$ and $\hat{r}_a^t$. On the other hand, by definition of $U_t = \{a \in [K] : \hat{r}_a^t \leq \frac{1}{(K-1)\epsilon}\}$, it holds that $\bar{r}^t \leq 1 + \frac{1}{(K-1)\epsilon}$. Then

$$\sum_{t=1}^T \sum_{a \notin U_t} x_a^t \big(\tilde{r}_a^t - \bar{r}^t\big)^2 \leq \sum_{t=1}^T \sum_{a \notin U} x_a^t \big(\tilde{r}_a^t\big)^2$$

since $\tilde{r}_a^t - \bar{r}^t \geq \hat{r}_a^t - \frac{1}{(K-1)\epsilon} \geq 0$ for each $a \notin U_t$ and $\bar{r}^t \geq 0$. Finally, for every $a \in U$, it holds that

$$\hat{r}_a^t \leq \frac{1}{(K-1)\epsilon}$$

since $x^t \in \mathrm{Conv}_\epsilon(\mathcal{A})$, and so $U \subseteq U_t$ for all time $t$. Substituting back in (16), we get

$$\mathbb{E}\left[\sum_{t=1}^T \big(v_\epsilon - v^t\big)^\top \big(\tilde{r}^t - \bar{r}^t \mathbf{1}\big)\right] \leq \frac{S \log(K/S)}{\eta}$$

$$+ \eta \mathbb{E}\Bigg[\underbrace{\sum_{t=1}^T \sum_{a \in U} x_a^t (1 - x_a^t)\big(\hat{r}_a^t\big)^2}_{(A)} + \underbrace{\sum_{t=1}^T \sum_{a \in U_t \backslash U} x_a^t (1 - x_a^t)\big(\hat{r}_a^t\big)^2}_{(B)} + \underbrace{\sum_{t=1}^T \sum_{a \notin U_t} x_a^t \big(\tilde{r}_a^t\big)^2}_{(C)}\Bigg].$$
(17)

First, we bound the expectation of term (A) as follows:

$$= \mathbb{E}\left[\sum_{t=1}^T \sum_{a \in U} x_a^t (1 - x_a^t)\big(\hat{r}_a^t\big)^2\right] = \mathbb{E}\left[\sum_{t=1}^T \sum_{a \in U} x_a^t \frac{\sum_{i \neq a} \mathbb{1}[v_i^t = 1](1 - r_a^t)}{1 - x_a^t}\right] \leq \mathbb{E}\left[\sum_{t=1}^T \sum_{a \in U} x_a^t \frac{\sum_{i \neq a} \mathbb{1}[v_i^t = 1]}{1 - x_a^t}\right]$$

$$= \mathbb{E}\left[\sum_{t=1}^T \sum_{a \in U} x_a^t\right] \leq ST.$$

Note (C) can be decomposed as follows:

$$\mathbb{E}\left[\sum_{t=1}^T \sum_{a \notin U_t} x_a^t \big(\tilde{r}_a^t\big)^2\right] = \mathbb{E}\left[\sum_{t=1}^T \sum_{a \notin U_t} x_a^t \big(1 - \hat{r}_a^t\big)^2\right] \leq \mathbb{E}\left[\sum_{t=1}^T \sum_{a \notin U_t} x_a^t \Big(1 + \big(\hat{r}_a^t\big)^2\Big)\right]$$

$$= \mathbb{E}\left[\sum_{t=1}^T \sum_{a \notin U_t} x_a^t\right] + \mathbb{E}\left[\sum_{t=1}^T \sum_{a \notin U_t} x_a^t \big(\hat{r}_a^t\big)^2\right].$$

Since $1 - x_a^t \in [0, 1]$ in term (B), we can plug the above bounds back in (17) and get

$$\mathbb{E}\left[\sum_{t=1}^T \big(v_\epsilon - v^t\big)^\top \big(\tilde{r}^t - \bar{r}^t \mathbf{1}\big)\right] \leq \frac{S \log(K/S)}{\eta} + ST + \mathbb{E}\left[\sum_{t=1}^T \sum_{a \notin U} x_a^t \big(\hat{r}_a^t\big)^2\right]$$

$$\leq \frac{S \log(K/S)}{\eta} + ST + T\left(S + 4\alpha \log\left(\frac{4KS}{\epsilon\alpha}\right)\right)$$

where the last inequality follows from (11). $\qquad \square$

## D. Arm elimination algorithm for stochastic rewards

As promised in Section 5.1, we present an elimination-based algorithm, called Combinatorial Arm Elimination, that is agnostic to the decision subset $\mathcal{A}_0$ and achieves regret $\widetilde{O}(S\sqrt{\alpha T})$. We assume the reward $r_i^t \in [0, 1]$ for each arm $i \in [K]$

---

**Algorithm 3** Combinatorial Arm Elimination

---

**Input:** time horizon $T$, decision subset $\mathcal{A}_0 \subseteq \mathcal{A}$, arm set $[K]$, combinatorial budget $S$, feedback graph $G$, and failure probability $\epsilon \in (0,1)$.

**Initialize:** Active set $\mathcal{A}_{\text{act}} \leftarrow \mathcal{A}_0$, minimum count $N \leftarrow 0$.

Let $(\bar{r}_a^t, n_a^t)$ be the empirical reward and the observation count of arm $a \in [K]$ at time $t$.

For each combinatorial decision $v \in \mathcal{A}_{\text{act}}$, let $\bar{r}_v^t = \sum_{a \in v} \bar{r}_a^t$ be the empirical reward and $n_v^t = \min_{a \in v} n_a^t$ be the minimum observation count.

**for** $t = 1$ **to** $T$ **do**

    Let $\mathcal{A}_N \leftarrow \{v \in \mathcal{A}_{\text{act}} : n_v^t = N\}$ be the decisions that have been observed least.

    Let $G_N$ be the graph $G$ restricted to the set $U_t = \{a \in [K] : \exists v \in \mathcal{A}_N \text{ with } a \in v\} = \bigcup_{v \in \mathcal{A}_N} v$.

    Let $a_t \in U_t$ be the arm with the largest out-degree (break tie arbitrarily).

    Pull any decision $v_t \in \mathcal{A}_N$ with $a_t \in v_t$.

    Observe the feedback $\{r_a^t : a \in N_{\text{out}}(v_t)\}$ and update $(\bar{r}_a^t, n_a^t)$ accordingly.

    **if** $\min_{v \in \mathcal{A}_N} n_v^t > N$ **then**

        Update the minimum count $N \leftarrow \min_{v \in \mathcal{A}_{\text{act}}} n_v^t$.

        Let $\bar{r}_{\max}^t \leftarrow \max_{v \in \mathcal{A}_{\text{act}}} \bar{r}_v^t$ be the maximum empirical reward in the active set.

        Update the active set as follows:

$$\mathcal{A}_{\text{act}} \leftarrow \left\{ v \in \mathcal{A}_{\text{act}} : \bar{r}_v^t \geq \bar{r}_{\max}^t - 6S\sqrt{\frac{\log(2T)\log(KT/\epsilon)}{N}} \right\}.$$

    **end**

**end**

---

is i.i.d. with a time-invariant mean $\mu_i$. The algorithm maintains an active set of the decisions and successively eliminates decisions that are statistically suboptimal. It crucially leverages a structured exploration within the active set $\mathcal{A}_{\text{act}}$. In the proof below and in Algorithm 3, for ease of notation, we let $v \in \mathcal{A}_0$ denote both the binary vector and the subset of $[K]$ it represents. So $a \in v \subseteq [K]$ if $v_a = 1$.

**Theorem D.1.** *Fix any failure probability $\epsilon \in (0,1)$. For any decision subset $\mathcal{A}_0 \subseteq \mathcal{A}$, with probability at least $1 - \epsilon$, Algorithm 3 achieves expected regret*

$$\mathbb{E}[\mathsf{R}(\text{Alg } 3)] = \widetilde{O}\left(S\alpha + S\sqrt{\log(KT/\epsilon)\alpha T}\right).$$

*Proof.* Fix any $\epsilon \in (0,1)$. For any $n \geq 0$, denote $\Delta_n = 3\sqrt{\log(2T)\log(KT/\epsilon)/n}$ (let $\Delta_0 = 1$ for simplicity). During the period of $N = n$, by Lemma F.6, with probability at least $1 - \epsilon$, we have $|\bar{r}_a^t - \mu_a| \leq \Delta_n$ for any individual arm $a \in U_t$ at any time $t$. In the remaining proof, we assume this event holds. Then the optimal combinatorial decision $v_*$ is not eliminated at the end of this period, since

$$\bar{r}_{v_*}^t \geq \mu_{v_*} - S\Delta_n \geq \mu_{\max} - S\Delta_n \geq \bar{r}_{\max}^t - 2S\Delta_n.$$

In addition, for any $v \in \mathcal{A}_{\text{act}}$, the elimination step guarantees that

$$\mu_v \geq \bar{r}_v^t - S\Delta_n \geq \bar{r}_{\max}^t - 3S\Delta_n \geq \bar{r}_{v_*}^t - 3S\Delta_n \geq \mu_{v_*} - 4S\Delta_n. \tag{18}$$

Let $T_n$ be the duration of $N = n$. Recall that $a_t \in U_t$ has the largest out-degree in the graph $G$ restricted to $U_t$. By Lemma F.1 and Lemma F.3, we are able to bound $T_n$:

$$T_n \leq (1 + \log(K))\delta(G_N) \leq 50\log(K)(1 + \log(K))\alpha(G_N) \leq 50\log(K)(1 + \log(K))\alpha \equiv M.$$

By (18), the regret incurred during $T_n$ is bounded by $4S\Delta_n T_n$. Thus with probability at least $1 - \epsilon$, the total regret is upper

bounded by

$$\mathbb{E}[\mathsf{R}(\text{Alg } 3)] \leq ST_0 + 4S \sum_{n=1}^{\infty} \Delta_n T_n$$

$$\leq SM + 4S \sum_{n=1}^{T/M} \Delta_n M$$

$$\leq SM + 12SM\sqrt{\log(2T)\log(KT/\epsilon)}\sqrt{T/M}$$

$$\leq SM + 12S\sqrt{\log(2T)\log(KT/\epsilon)}\sqrt{MT}$$

$$\leq SM + 60\sqrt{\log(K)(1+\log(K))\log(2T)\log(KT/\epsilon)}S\sqrt{\alpha T}.$$

$\square$

## E. Proof of Theorem 5.2

The proof of Theorem 5.2 follows that of Theorem 3.2. The only difference is that the correlation condition of $p^t$ is no longer guaranteed on general $\mathcal{A}_0$. Now we can only bound (10) as $\mathbb{E}\left[\left(\sum_{i\in N_{\text{in}}(a)} v_i^t\right)^2\right] \leq S\mathbb{E}\left[\sum_{i\in N_{\text{in}}(a)} v_i^t\right]$. Then (11) becomes

$$\sum_{t=1}^{T}\sum_{a\notin U} x_a^t \frac{\mathbb{E}\left[\left(\sum_{i\in N_{\text{in}}(a)} v_i^t\right)^2\right]}{\left(\sum_{i\in N_{\text{in}}(a)} x_i^t\right)^2} \overset{(a)}{\leq} \sum_{t=1}^{T}\sum_{a\notin U} x_a^t \frac{S\mathbb{E}\left[\sum_{i\in N_{\text{in}}(a)} v_i^t\right]}{\left(\sum_{i\in N_{\text{in}}(a)} x_i^t\right)^2}$$

$$= \sum_{t=1}^{T}\sum_{a\notin U} x_a^t \frac{S\sum_{i\in N_{\text{in}}(a)} x_i^t}{\left(\sum_{i\in N_{\text{in}}(a)} x_i^t\right)^2}$$

$$= \sum_{t=1}^{T}\sum_{a\notin U} S\frac{x_a^t}{\sum_{i\in N_{\text{in}}(a)} x_i^t}$$

$$\leq \sum_{t=1}^{T}\sum_{a\notin U} S\frac{x_a^t}{\sum_{i\notin U:i\in N_{\text{in}}(a)} x_i^t}$$

$$\overset{(b)}{\leq} 4S\alpha T\log\left(\frac{4K}{\alpha\epsilon}\right)$$

where (a) is by $\|v^t\|_1 \leq S$ and (b) uses Lemma F.2. Plugging this back to (11) in the proof of Theorem 3.2 yields the first bound. When the feedback graphs are time-varying, one gets instead $\widetilde{O}\left(S\sqrt{\sum_{t=1}^{T}\alpha_t}\right)$.

## F. Auxiliary lemmas

For any directed graph $G = (V, E)$, one can find a dominating set by recursively picking the node with the largest out-degree (break tie arbitrarily) and removing its neighbors. The size of such dominating set is bounded by the following lemma:

**Lemma F.1** ((Chvatal, 1979)). *For any graph $G = (V, E)$, the above greedy procedure outputs a dominating set $D$ with*

$$|D| \leq (1 + \log|V|)\delta(G).$$

**Lemma F.2** (Lemma 5 in (Alon et al., 2015)). *Let $G = ([K], E)$ be a directed graph with $i \in N_{\text{out}}(i)$ for all $i \in [K]$. Let $w_i$ be positive weights such that $w_i \geq \epsilon \sum_{i\in[K]} w_i$ for all $i \in [K]$ for some constant $\epsilon \in (0, \frac{1}{2})$. Then*

$$\sum_{i\in[K]} \frac{w_i}{\sum_{j\in[K]:j\to i} w_j} \leq 4\alpha\log\left(\frac{4K}{\alpha\epsilon}\right)$$

**Lemma F.3** (Lemma 8 in (Alon et al., 2015)). *For any directed graph $G = (V, E)$, one has $\delta(G) \leq 50\alpha(G) \log |V|$.*

**Lemma F.4.** *Let $F : X \to \mathbb{R}$ be a convex, differentiable function and $D \subset \mathbb{R}^d$ be an open convex subset. Let $x_* = \arg\min_{x \in D} F(x)$. Then for any $y \in D$, $(y - x_*)^T \nabla F(x_*) \geq 0$.*

*Proof.* We will prove by contradiction. Suppose there is $y \in D$ with $(y - x_*)^T \nabla F(x_*) < 0$. Let $z(t) = F(x_* + t(y - x_*))$ for $t \in [0, 1]$ be the line segment from $F(x_*)$ to $F(y)$. We have

$$z'(t) = (y - x_*)^T \nabla F(x_* + t(y - x_*))$$

and hence $z'(0) = (y - x_*)^T \nabla F(x_*) < 0$. Since $D$ is open and $F$ is continuous, there exists $t > 0$ small enough such that $z(t) < z(0) = F(x_*)$, which yields a contradiction. $\square$

**Lemma F.5** (Chapter 11 in (Cesa-Bianchi & Lugosi, 2006)). *Let $F$ be a Legendre function on open convex set $\mathcal{D} \subseteq \mathbb{R}^d$. Then $F^{**} = F$ and $\nabla F^* = (\nabla F)^{-1}$. Also for any $x, y \in \mathcal{D}$,*

$$D_F(x, y) = D_{F^*}(\nabla F(y), \nabla F(x)).$$

**Lemma F.6** (Lemma 1 in (Han et al., 2024)). *Fix any $\epsilon \in (0, 1)$. With probability at least $1 - \epsilon$, it holds that*

$$\left| \bar{r}_a^t - \mu_a \right| \leq 3\sqrt{\frac{\log(2T) \log(KT/\epsilon)}{n_a^t}}$$

*for all $a \in [K]$ and all $t \in [T]$.*

