# OpenReview forum: "Adversarial Combinatorial Semi-bandits with Graph Feedback"
_ICML.cc/2025/Conference — ICML 2025 poster_

### Official Review · Reviewer_R6q4 · 2025-02-25

**Overall Recommendation:** 3

**Summary:**

**Edit post-rebuttal: I thank the authors for their feedback, which answered my questions. I maintain my overall positive score.**

The submission considers adversarial combinatorial semi-bandits, with additional feedback, ranging from no additional feedback to full-information feedback. The two extreme cases were respectively studied by Audibert, Bubeck and Lugosi (2014, no additional feedback) and Koolen, Warmuth, Kivinen, (2010, full-information feedback). The corresponding optimal regret bounds (up to logarithmic factors) are respectively $\sqrt{K S T}$ and $S \sqrt{T}$, where $S$ is the number of arms played at each round and $K$ is the total number of arms. The present submission interpolates between these two extremes (both in terms of upper and lower bounds), by suggesting a model of graph-determined feedback and by stating optimal regret bounds in terms of the richness of the graph (quantified by the independence number of the graph). The two extremes are given by a graph with inner loops at each node, and no other edge, and by the complete graph. Note that the regret upper bounds also work for a sequence of adversarial feedback graphs.

**Claims And Evidence:**

The results are theoretical: I could not spot flaws in the proofs---though their clarity could sometimes be improved, and though large parts of the proofs could be made more concise, by using existing results; see details below.

**Essential References Not Discussed:**

The submission discusses all references I expected to find.

**Experimental Designs Or Analyses:**

N/A

**Methods And Evaluation Criteria:**

The setting considered makes sense. The regret is only handled in expectation, and I think that high-probability bounds would have been appreciated.

**Other Comments Or Suggestions:**

- Page 1, column 1, line 97: I guess this should be $\mathbb{E}[p_a^t]$ and not $\mathbb{E}[v_a^t]$?
- Page 1, column 2, line 64: put the definition of $\alpha$ and $\delta$ before using them
- Page 2, Lemma 1.2: do we agree that, in some sense, Conv($A$) and the probability distributions over $A$ are the same set?
- Page 2, Section 1.3: the concepts of independent subset and dominating subset should be defined, to make the submission self-complete
- Page 8: $\delta$ is only used at this stage, its definition could thus be postponed here

**Other Strengths And Weaknesses:**

This submission provides an interesting, elegant, and optimal interpolation between existing optimal regret bounds for combinatorial semi-bandits. Most of the proof techniques were known (lower bound proof structure; pre-regret upper bound in terms of the estimates; Lemma D.2 by Alon et al., 2015) but they are well assembled and there seems to be a new ingredient provided by the representation of Lemma 1.2.

On a side note, I particularly appreciated the comments and insights throughout the text; e.g., page 3, column 2, lines 117-125, or page 6, column 1, lines 282-295

**Questions For Authors:**

I have no specific question but feedback from the authors on some points raised above (e.g., whether high-probability bounds are easy to obtain; whether indeed the application of Theorem 3 by Audibert, Bubeck, Lugosi, 2014, would indeed save 1 page, etc.) are welcome.

**Edit post-rebuttal: I thank the authors for their feedback, which answered my questions. I maintain my overall positive score.**

**Relation To Broader Scientific Literature:**

N/A

**Theoretical Claims:**

I checked in details the main two results: Theorem 1.1 and Theorem 1.3 (based on Lemma 1.2, proved in Appendix A, whose proof I did not check in detail).  I could generally follow the derivations.

Theorem 1.1
- Could you detail (page 4, column 1, line 182) why this assumption comes with no loss of generality? It would look more satisfactory to me to rather have some of the $I_j$ contain $n$ elements and some others to contain $n-1$ elements, and write the proof in this context
- Page 4, first two thirds of column 2: the arguments could be made more compact (and would avoir resorting to densities) by using the bound by Ménard, Garivier, and Stoltz (2019)
- Page 4, Equation (4): first, $T_0$ is not defined (it is the number of pulls in $I$); second, I think that this is the place where you critically use the definition of $I$ as a maximal independent set and this must be detailed and commented. This is the crux of the proof, while earlier manipulations were standard
- The remarks stated right before Section 3 are nice and should have been stated before the formal proof, as they help navigating it

Theorem 1.3
- The beginning of this proof (possibly from page 6, column 1, line 317 to page 7, column 1) could be replaced by an appeal to the bound of Theorem 3 of Audibert, Bubeck and Lugosi (2014), which actually holds for all estimators; i.e., the new part consists of the end of the proof, where, in particular, the negativity of the covariance of the $p^t$ is used

---

> ### Author Rebuttal · Authors · 2025-03-30
>
> The detailed review and insightful feedback from the reviewer are deeply appreciated. We have done several updates in light of the review in our revision.
> - It is worth mentioning that our previous lower bound construction did not lead to the desired trade-off and was wrong. We corrected it by constructing $S$ independent sub-problems via multi-index $u\in[n]^S$ (instead of $u\in[n]$) and adopted a stopping time argument from [1]. The crux of the lower bound is that the learner has to learn each of the $S$ sub-problems well and independently. As an intuitive reasoning, a good learner tends to pick one arm of each sub-problem, i.e. distributes the pulls roughly evenly, because if he/she picks two, then one of the two must be suboptimal and incurs regret. This new proof will be deferred to the appendix due to space limit and that it is not the particular contribution of this work. We will provide an intuitive sketch of the proof in Section 2 that includes the remark mentioned by the reviewer at the beginning.
> - Regarding the comments on the lower bound: we may assume $\alpha > 3S$, since otherwise $S\sqrt{T}=\Omega(\sqrt{\alpha ST})$ and we are done. Then we only need to consider the independent subset with size $\lfloor \alpha/S\rfloor S=nS\le alpha$. Note since $\alpha > 3S$, it holds that $nS\ge \alpha-S\ge \alpha/2$, so the final bound would be the same up to a factor of 2.
> - We apologize for the confusion. $T_0$ was a typo and was meant to be $N_0$, which counts the number of pulls outside of the independent subset $I$. So the regret would be $\Omega(N_0)$ since any node outside (while they may be more informative) incurs a constant regret.
> - We agree that the beginning of the upper bound proof overlaps with that of the earlier work. Nonetheless, it was included for the sake of rigor because we use an additional truncation on the convex hull.
> - The high probability upper bound may be a nontrivial extension and we will leave it to future work.
> - The comments are carefully addressed in our revision. In particular, there is a surjective mapping from $Conv(\mathcal{A})$ to the probability distributions over $\mathcal{A}$, so in this sense it is easy to obtain a distribution with mean $x\in Conv(\mathcal{A})$. But the requirement of negative correlations in our case is nontrivial to satisfy and, as shown in Section 4.1, is impossible for some $\mathcal{A}$.
>
> References:
>
> [1] Lattimore, Tor and Kveton, Branislav and Li, Shuai and Szepesvari, Csaba. TopRank: A Practical Algorithm for Online Stochastic Ranking, NeurIPS 2018.

---

### Official Review · Reviewer_TtKB · 2025-03-02

**Overall Recommendation:** 4

**Summary:**

The authors consider the problem of combinatorial semi-bandit with feedback graphs, where a graph over the $K$ actions may provide the learner with side information during the learning process. By presenting appropriate lower and upper bounds, the authors establish a minimax optimal regret bound for graphs containing self-loops of $\widetilde \Theta(S \sqrt{T} + \sqrt{\alpha S T})$, where $S$ is the size of a combinatorial action and $\alpha$ is the independence number of the underlying feedback graph. For the upper bound, the authors design a natural extension of the OSMD algorithm to the feedback graph setting, with the crucial property that the sampling distributions $p_t$ satisfy a negative correlation property, which allows obtaining the optimal dependence on $S$ for this algorithm. Furthermore, the authors consider scenarios where not all combinatorial actions belong to the available action set, and show that in such settings it may not be possible to satisfy the negative correlation property, and in fact the optimal rates in such settings are worse by a $\sqrt{S}$ factor. The authors also briefly discuss the weakly observable setting and analyze an Explore-Then-Exploit algorithm for this setting with stochastic rewards.

## update after rebuttal:
Given the latest response by the authors, they seem to have a reasonably convincing argument showing that if negative correlations are not explicitly enforced in OSMD, the algorithm could incur suboptimal regret. Given the validity of their argument, I think that such a result significantly strengthens the contributions of the paper, and I highly encourage the authors to include it in the final version. I am now inclined to increase my score to "Accept".

**Claims And Evidence:**

The authors provide formal proofs for all of their results, contributions and technical claims. The analysis in the paper is clear and easy to follow, with many parts being very similar to analysis from previous related work.

**Essential References Not Discussed:**

When discussing full-bandit feedback, the authors do not mention previous works that prove that the optimal bounds are of the form $\Theta(S^{3/2} \sqrt{KT})$, shown in [1] for a specific combinatorial action set (namely, multitask bandits) and in [2] for the full action set. While the full-bandit variant is not extremely relevant to this work, since the authors mention this variant and cite some related work, they should also mention the aforementioned papers in which the optimal rates are characterized.

References:

[1] Cohen, Alon, Tamir Hazan, and Tomer Koren. "Tight bounds for bandit combinatorial optimization." Conference on Learning Theory. PMLR, 2017.

[2] Ito, Shinji, et al. "Improved regret bounds for bandit combinatorial optimization." Advances in Neural Information Processing Systems 32 (2019).

**Experimental Designs Or Analyses:**

N/A

**Methods And Evaluation Criteria:**

N/A

**Other Comments Or Suggestions:**

* I suggest that the authors further highlight their main technical novelty regarding the negatively correlated arms, as it seems highly nontrivial and without it the rest of the techniques are pretty straightforward extensions from previous works. On that regard, I think that less focus should be given to the lower bound in the main text, as it seems pretty straightforward and doesn't include any highly novel techniques. Instead, the authors could provide further intuition and discussion on the negative correlation property, specifically, I would appreciate a high level intuition of why there must exist such a distribution, as the proof of Lemma 1.2 is quite technical and not so intuitive.

**Other Strengths And Weaknesses:**

Strengths:

* The authors study a well-motivated problem which has not been studied in previous works, namely combinatorial semi-bandits with feedback graphs.
* The authors establish optimal (up to logarithmic factors) regret bounds for this problem under mild assumptions on the feedback graph.
* The analysis presented by the authors is clear, easy to follow, and in large parts resembles analysis of similar problems from previous works.
* The key technical observation made by the authors, namely, that using sampling distributions with negative correlation between arms is essential in order to obtain optimal bounds, seems very interesting to me, and I'm not sure that I've seen anything similar in previous works. The fact that such sampling distributions exist also seems nontrivial.
* The authors support the fact that the negative correlation property is crucial by exhibiting instances with a limited decision set in which this property doesn't hold, and the optimal regret bound is in fact worse by a factor of $\sqrt{S}$.

Weaknesses:
* It seems to me that other than the key observation regarding the negatively correlated arms (specifically, Lemma 1.2 and how it is used in the proof of Theorem 3.2), the other parts of the analysis (including the lower bound) uses fairly standard techniques which are very common in related previous work. Therefore, I think perhaps this key observation is not highlighted well enough in the current version of the paper, and it is a bit lost among the other parts of the analysis which are fairly straightforward.
* The authors mention that extending their results from graphs with self-loops to strongly observable graphs is straightforward without any elaboration. However, in previous works, specifically in Alon et al. ('15), such an extension requires some additional techniques which are not present for graphs with self-loops. I suggest that the authors either elaborate on how such an extension can be performed for their results, or alternatively remove this remark altogether.

**Questions For Authors:**

1. I would appreciate it if the authors could comment on the possible issue I mentioned under "Theoretical Claims".

2. Do the authors have an example where if their algorithm OSMD.G is run without the negative correlation condition, then this algorithm can incur regret of the form $\Omega(S \sqrt{\alpha T})$ even when the action set is full? I think that such an example can considerably strengthen the results, as it would imply that explicitly requiring this condition is essential for this algorithm. Given such a result, I may consider increasing my evaluation score of the paper.

3. Regarding the weakly-observable case, have the authors attempted an extension of EXP3.G (Alon et al., '15) to the combinatorial setting? If so, can the authors comment on why such an extension is nontrivial?

**Relation To Broader Scientific Literature:**

The authors discuss relevant works from both the bandits with feedback graphs literature and combinatorial semi-bandit literature. Specifically, they cite Alon et al. ('15) inwhere optimal regret bounds are established for the non-combinatiral variant of the problem, and Audibert et al. ('14) who presented the OSMD algorithm which achieves optimal regret bounds for combinatorial semi-bandits with no feedback graphs.

**Theoretical Claims:**

I went over some of the proofs of the theoretical claims in the paper, and found one possible issue which probably does not affect the overall claim, but requires the authors' attention. Specifically, in the upper bound analysis, when bounding the second order term, the authors bound a term of the form $\sum_a \frac{x^t_a}{\sum_{i \to a} x^t_i}$ by an order of $\alpha$ by referring to Lemma 5 of [1]. However, examining the conditions of this lemma, using it as is requires the sum $\sum_a x^t_a$ to be bounded by 1, while in the combinatorial semi-bandit setting it is in fact bounded by $S$. This can be mitigated if the authors modify the optimization domain $Conv_{\epsilon}(\mathcal{A})$ to $Conv_{S \epsilon}(\mathcal{A})$ so that the iterates would satisfy $x^t_a \geq \epsilon \sum_i x^t_i$ as required from the conditions of the lemma. This modification would result in a slightly worse dependence on $\epsilon$ in the overall bound, which will ultimately affect the additive term ($\epsilon SKT$ instead of $\epsilon K T$) and the logarithmic term, and thus would not change the overall claim and contributions. I would appreciate it if the authors could address this issue.

Reference:

[1] Alon, Noga, et al. "Online learning with feedback graphs: Beyond bandits." Conference on Learning Theory. PMLR, 2015.

---

> ### Author Rebuttal · Authors · 2025-03-30
>
> We sincerely appreciate the reviewer's thorough review and comments and have done several updates in our revision in light of the review.
> - We appreciate the review spotting the issue of missing a factor $S$ in the log term. We have corrected this in our revision.
> - We agree that the lower bound is not the focus of this work (in fact, its proof was wrong as pointed out by Reviewer bn5t; we have corrected the hard instance construction and proof techniques in our revision). We will move the proof to the appendix and only briefly discuss the construction in the main text, in order to keep our focus on the upper bound ideas. Instead, we provide an intuitive reasoning for why such distributions exist, which is essentially the high-level idea of our proof of Lemma 1.2. Since the revision cannot be posted, we will paste this reasoning here for clarity:
>
> When $S=1$, any distributions possess negative correlations. Inductively, let us suppose such distributions exist for $1,2,\dots, S-1$. Then for a fixed target $x\in\mathsf{Conv}(\mathcal{A})$, we can always find an index $i\in[K]$ such that $\sum_{j=1}^{i-1}x_j + cx_i=1$ and $\sum_{j=i+1}^Kx_j + (1-c)x_i=S-1$ for some $c\in[0,1]$. Namely, the target of size $S$ is partitioned into two sub-targets with ranges $[1,i]$ and $[i,K]$, each with sizes $1$ and $S-1$, and with an overlap on index $i$. We can then assign $v_i=0$ with probability $1-x_i$, to the first half $[1,i]$ with probability $cx_i$, and to $[i,K]$ with probability $(1-c)x_i$. To obtain a final size $S$ solution, we draw $v'$ supported on $[1,i-1]$ with size $0$ or $1$ and $v''$ on $[i+1,K]$ with size $S-1$ or $S-2$, conditioned on the assignment of $v_i$. For any $j_1\in [1,i-1]$, $j_2\in[i+1,K]$, and $i$, any two of them are negatively correlated because, at a high level, the presence of one `reduces' the budget size of the other. The negative correlations among the first half $[1,i-1]$ and $[i+1,K]$ are guaranteed by the induction hypothesis of the existence of such distributions for solutions with size less than $S$. Finally, the structure of $\mathcal{A}$ ensures that our pieced-together solution is valid, i.e. lies in $\mathcal{A}$.
>
> - We have removed the assumption on self-loops in $G$ and extended our results to general strongly observable graphs. The extension is done via an adaptation of the efforts in [1] to our reward setting. In the discussion, we will also generalize $\widetilde{\Theta}(S\sqrt{T}+\sqrt{\alpha ST})$ to a subclass of decision sets (including the full set) satisfying this exchange property: for any $u,v\in\mathcal{A}_0$, there exist $i\in u$ and $j\in v$ such that $u-i+j$ and $v-j+i$ remain in $\mathcal{A}_0$. Examples include that a learner operates $S$ systems in parallel and chooses one action for each system at each time, such as multi-platform online advertising.
> - We agree that a comparison between the algorithms with/without negative correlations will be interesting and have practical value. Our focus is nonetheless on the theoretical understanding on the regret characterization, and thus we have not conducted numerical comparisons. Theoretical analyses of the two algorithms, on the other hand, require precise understandings of the trajectories of $x^t$ in OSMD-G and may be considered in future work.
> - It is definitely possible to adapt EXP3.G to the combinatorial setting for the weakly observable graphs. However, we believe the importance lies in figuring out the lower bound in the combinatorial setting (under full decision set or proper subsets). Straightforward analysis can give bounds like the one we have for ETC in Section 4.2, but they are interesting only if they turn out to be tight. In particular, in our study of the strongly observable case, preliminary results gave the upper bound $\widetilde{O}(S\sqrt{\alpha T})$ and the lower bound $\Omega(S\sqrt{T} + \sqrt{\alpha ST})$ and we did not know which one was tight. It then turns out they are both tight under different decision set structures. So the interesting question, rather than generalizing EXP3.G, would be the tight characterization under weakly observable graphs, which may again depend on the decision set. We do not have a lower bound beyond the existing $\Omega(\delta^{1/3}T^{2/3})$ for now.
>
> Again, we would like to thank the reviewer for the helpful feedback, and we are more than happy to answer any questions.
>
> References:
>
> [1] Alon, Noga and Cesa-Bianchi, Nicolo and Dekel, Ofer and Koren, Tomer. Online learning with feedback graphs: Beyond bandits. COLT 2015.

---

> > ### Comment · Reviewer_TtKB · 2025-04-02
> >
> > I thank the authors for their response.
> >
> > Regarding my second question for the authors, I was not referring to numerical analysis, but rather to a theoretical result possibly establishing an algorithm-specific lower bound of a vanilla version of their algorithm (one which doesn't guarantee negative correlations in the sampling distributions). If the authors can prove that such a variant has a suboptimal regret lower bound, I think it significantly strengthens their result, and would be a very nice addition to this paper as it currently seems that the main technical novelty is this negative correlation property.
> >
> > Having read the authors' rebuttal, I am currently inclined to maintain my score.

---

> > > ### Author Response · Authors · 2025-04-02
> > >
> > > We thank the reviewer's reply and clarification. In fact, we realized it is *possible* to construct a hard instance such that OSMD with positive correlations suffers $\Omega(S\sqrt{\alpha T})$, which gives a good complement to our story. The statement and the idea are as follows:
> > >
> > > > Consider the full decision set and given $(K,S,\alpha,T)$. There is a graph $G$ with $\alpha(G)=\alpha$, a choice of mapping $F$, and a sampling scheme $p^t$ with which the OSMD has a minimax regret lower bound $\Omega(S\sqrt{\alpha T})$, when $S\alpha \le K$ and $ST\ge \alpha^3$.
> > >
> > > We consider the negative entropy mirror mapping $F$ (this can be further relaxed). The vanilla OSMD only requires $p^t$ to satisfy the mean condition, denoted by **(M)**, and we now choose a type of $p^t$ satisfying **(M)** but leading to positive correlations, and we construct an instance in which this type of OSMD suffers $\Omega(S\sqrt{\alpha T})$. To save space, consider the feedback graph $G$ and $H$ and the partition in Section 4.1, with cliques $V_1,...,V_{K/S}$ each with size $S$. Let $p^t$ be the following:
> > > - (1). If the target $x^t$ has same values over each clique, i.e. for each $V_i$ and any $u,v\in V_i$, $x^t_u=x^t_v\equiv x^t(V_i)$, then $p^t$ draws $V_i$ with probability $x^t(V_i)$ (note this is a valid distribution, since $S = \sum_{u\in[K]}x^t_k = S\sum_{i\in[K/S]}x^t(V_i)$).
> > > - (2). Otherwise use any $p^t$ satisfying **(M)**.
> > >
> > > Key observation is, if the rewards $r^t$ also has same values over each clique (let's call this property **(P)**), we can show (2) never happens and thus $p^t$ in (1) always has positive correlations. This claim can be seen in a few steps and an inductive argument:
> > > - The uniform initialization $x^1=\frac{1}{K}\mathbf{1}$ (given by minimizing the negative entropy) satisfies **(P)**.
> > > - Then $p^t$ draws $v^t$ as defined in (1).
> > > - Since $r^t$ satisfies **(P)** and the feedback given by $G$ reveals either another clique in entirety or none, the constructed reward estimator $\tilde{r}^t$ satisfies **(P)**.
> > > - Then the solution $w^{t+1}$ satisfies **(P)** by the update rule in eq.(6); and the projected solution $x^{t+1}$ satisfies **(P)** under KL projection onto the truncated convex hull.
> > >
> > > Inductively, $p^t$ is given by (1) for every $t$. So when the rewards $r^t$ respects **(P)**, this OSMD reduces to an MAB algorithm running on the cliques $V_1,...,V_{K/S}$ with graph $H$ (similar to the construction in Section 4.1). From the MAB lower bound, we know there is a set of hard instances with rewards $R^t$ that leads to $\Omega(S\sqrt{\alpha T})$, where $R^t_i$ is the *group* reward of the clique $V_i$ in the MAB lower bound. We just split it equally over the clique to obtain $r^t$, namely $r^t_u=\frac{R^t_i}{S}$ for $u\in V_i$, that respects property **(P)** and thus leads to the aforementioned behavior and the desired lower bound.
> > >
> > > **Note** that the previous work [1] proves bounds for OSMD with negative entropy $F$ and *any* $p^t$ as long as **(M)** is satisfied. This counterexample now shows the structure of $p^t$ is crucial to leverage the presence of additional feedback, so **(M)** is not enough.
> > >
> > > [1] Audibert, J.-Y., Bubeck, S., and Lugosi, G. Regret in online combinatorial optimization.

---

### Official Review · Reviewer_bn5t · 2025-03-14

**Overall Recommendation:** 3

**Summary:**

The paper extends the standard combinatorial semi-bandit problem by incorporating a feedback graph $G$ that allows the learner to observe rewards not just from the arms selected in the combinatorial action but also from their neighbors in $G$.
The authors show that the optimal regret scales as $S\\sqrt{T} + \\sqrt{\\alpha ST}$ ignoring logarithmic factors, where $S$ is the combinatorial action size, $\\alpha$ is the independence number of $G$, and $T$ is the number of rounds.
This result interpolates between the full-information and semi-bandit feedback scenarios.
To achieve this regret guarantee, the authors propose an algorithm called OSMD-G that adopts online stochastic mirror descent (OSMD) with unbiased reward estimators for graph feedback and sampling distributions that ensure negative correlations among distinct arms.
The authors also provide a regret lower bound of the same order.

**Claims And Evidence:**

Most claims in the paper are supported.
However, some are stated without providing enough intuition as to why they would be correct.
For instance, stating that the results assuming all self-loops in $G$ immediately extend to all strongly observable feedback graphs (e.g., see line 117) is not necessarily true.
Even for standard (non-combinatorial) bandits with feedback graphs, extending the analysis to general strongly observable feedback graphs is definitely nontrivial and requires specialized technical results for it (Alon et al., 2015).
One may therefore be convinced that it is even more nontrivial to have an analogous extension in the combinatorial setting.
Moreover, throughout the paper the authors often overly generalize their claims by stating they hold for "general directed feedback graphs" while, again, they make the assumption of including all self-loops in the graphs considered.

**Essential References Not Discussed:**

Most of the relevant references are discussed.
Other meaningful references on combinatorial bandits to include could be:
- Richard Combes, Mohammad Sadegh Talebi, Alexandre Proutiere, Marc Lelarge. Combinatorial Bandits Revisited. NeurIPS 2015.
- Alon Cohen, Tamir Hazan, Tomer Koren. Tight Bounds for Bandit Combinatorial Optimization. COLT 2017.
- Lukas Zierahn, Dirk van der Hoeven, Nicolò Cesa-Bianchi, Gergely Neu. Nonstochastic Contextual Combinatorial Bandits. AISTATS 2023.

**Experimental Designs Or Analyses:**

N/A

**Methods And Evaluation Criteria:**

N/A

**Other Comments Or Suggestions:**

- It is good practice to avoid using a parenthesized citation when it is a subject/object of a sentence (e.g., Koolen et al. (2010) instead of (Koolen et al., 2010) in line 91).
- Another good practice is to place footnote numbers after eventual punctuation (e.g., line 118).
- A more standard notation for a directed edge uses an ordered pair $(a,i)$ instead of $a \\to i$, as $E$ is a subset of ordered pairs of nodes (e.g., in line 74).
- The self-loops assumption is actually a crucial and important assumption, but is only mentioned as footnotes (namely, footnotes 1 and 2). It would be better to state it in the main body.
- The type of feedback for which the learner only observed the realized payoff $\\langle v^t, r^t\\rangle$ is often called "full-bandit" feedback.
- The algorithm from Alon et al. (2015), called Exp3.G, is described as an explore-then-commit algorithm while it is not (see line 71).
- Line 75: time-varying feedback graphs are also studied in Alon et al. (2015).
- In the statement of Theorem 1.1, underline that it holds for any directed graph $G$ containing all self-loops instead of an arbitrary one.
- $\\overline D$ (with an overline) is used without being defined. Please, provide a definition for it.
- In line 243, it would be clearer to specify that (c) holds for the concavity of the square root together with, e.g., Jensen's inequality.
- The log factors at lines 336, 383, and 675, following from the application of Lemma D.2, is missing an $S$ multiplicative factor in its argument.
- The square power at the expectation at lines 347, 358, and 665 should be inside the square brackets.

Typos:
- The multiplicative factor $T$ in the regret bound at line 78 should not be there.
- Line 147: $\\mathcal{A}$ instead of $A$.
- Line 198: $\\Omega$ should actually be $[n]$, right?
- Line 190: $a\_{m,u}$ instead of $a\_{m,u\_m}$.
- Lines 252 and 277: occurrences of $x$ are missing a superscript $t$.
- Line 274: $\\mathbb{E}[v] = x^t$ is missing a subscript $v \\sim p^t$, as otherwise even $x^t$ is a random variable (as similarly in multiple other places in this paper).
- Line 348: same as line 274, but with a missing outer expectation too.
- In the statement of Lemma A.1, quantify $S$ before using it in the definition of $\\mathcal{A}$.
- Line 517: "basis vector" instead of "unit vector".
- Line 528: $i>r$ instead of $i \\ge r$.

**Other Strengths And Weaknesses:**

### Post-rebuttal edit:

I thank the authors for providing detailed responses to address my concerns about the contents of their submission. I am satisfied with the responses. Provided the authors can guarantee to integrate all the modifications required to correct the issues and address the concerns pointed out in my review and the above discussion, I now have no strong reservations about the acceptance of this paper and I have updated my score accordingly.

To respond to the authors' final question about prior work without assuming knowledge of the independence number, here are a few relevant references:
- Kocák, Neu, Valko, and Munos. "Efficient learning by implicit exploration in bandit problems with side observations". NeurIPS 2014.
- Alon, Cesa-Bianchi, Gentile, Mannor, Mansour, Shamir. "Nonstochastic Multi-Armed Bandits with Graph-Structured Feedback". SIAM Journal on Computing, 2017.
- Rouyer, Van der Hoeven, Cesa-Bianchi, and Seldin. "A Near-Optimal Best-of-Both-Worlds Algorithm for Online Learning with Feedback Graphs". NeurIPS 2022.
- Eldowa, Esposito, Cesari, Cesa-Bianchi. "On the Minimax Regret for Online Learning with Feedback Graphs". NeurIPS 2023.

**Questions For Authors:**

- Please, address the main concerns pointed out above, if possible.
- A minor question: could you be more precise on what you mean by "tabular" contextual bandits (line 81)?

**Relation To Broader Scientific Literature:**

The regret bound for OSMD-G follows directly from the known analysis of OSMD combined with the variance bound for feedback graphs.
The only technical contribution here is the design of the distribution $p^t$ to allow a reduction to the standard analysis for the variance with feedback graphs.
The applicability of the results also requires quite restrictive assumption that are commonly lifted in the related literature, such as the presence of all self-loops in the strongly observable feedback graph, the knowledge of the independence number (NP-hard to compute) for tuning the learning rate, and considering only the full combinatorial action space $\\mathcal{A}$.
All of these point seem to underline a rather limited contribution of this submission.

**Theoretical Claims:**

I examined all the theoretical claims of the submission.
First, I would focus on the proof of Theorem 1.3, i.e., the regret lower bound.
Checking the end of the proof, there seems to be some issue with the computation.
Line 248 states that the regret $R(\\pi)$ of an algorithm $\\pi$ satisfies
$$
\\max\_{u \\in [n]} \\mathbb{E}\_u[R(\\pi)]
\\ge \\Delta ST (1/2 - 8\\Delta S\\sqrt{T/(3\\alpha)})
$$
whose right-hand side, replacing for the chosen value of $\\Delta$, becomes $\\frac{1}{32}\\sqrt{\\alpha ST} (1/2 - \\sqrt{S/48})$ which is non-positive, and thus vacuous, for $S \\ge 12$.

Additionally, even before that, line 194 shows a sum where each term presents a KL-divergence $\\mathrm{KL}(\\mathbb{P}_0(r^t|R^{t-1}) \\| \\mathbb{P}_u(r^t|R^{t-1}))$.
The issue here is that $\\mathbb{P}_0(r^t|R^{t-1})$ and $\\mathbb{P}_u(r^t|R^{t-1})$ present the entire reward vector $r^t$, and thus their KL-divergence is equal to $S \\cdot \\mathrm{KL}(\\mathrm{Bern}(1/4) \\| \\mathrm{Bern}(1/4+\\Delta))$ (by the chain rule over the coordinates of $r^t$), instead of only over $m \\in [S]$ such that $a\_{m,u} \\in N\_{\\mathrm{out}}(v^t)$.
Nonetheless, it is possible that a more careful analysis might avoid this issue.

Other that the proof of the lower bound, the other results seem to hold.

---

> ### Author Rebuttal · Authors · 2025-03-30
>
> We sincerely appreciate the reviewer's detailed comments and, most importantly, for pointing out our error in the lower bound. We have the following updates in our revision in light of the review:
> - We have corrected the lower bound. Specifically, we now construct $S$ independent sub-problems by an $S$ dimension index $u\in[n]^S$ (while previously they were correlated through single index $u\in[n]$) and apply the stopping time argument in Theorem 2 of [1] to establish the exploration-exploitation tradeoff for each sub-problem. Regarding the KL chain rule, it is crucial (as in many previous works, e.g. [1, 2]) that our quantity of interest only depends on the observed rewards, so the distribution of the $m$-th sub-problem only differs when the node $a_{m,u_m}$ is observed. This refinement leads to the bound $\sum_{m=1}^S\mathbf{1}[a_{m,u_m}\in N_{out}(v^t)]\cdot \Delta^2$ instead of $S\cdot \Delta^2$.
> - We have removed the assumption on self-loops for $G$ and extended our results to general strongly observable graphs. The extension is done via an adaptation of the efforts in [3] to our reward setting. Additionally, we have clarified our claim statements to specify that they apply to strongly observable graphs rather than arbitrary graphs.
> - To address concerns about the practical applicability of our results, we will add a section to explain how $\widetilde{O}(S\sqrt{T} + \sqrt{\alpha ST})$ applies to a subclass of decision sets satisfying this exchange property: for any $v,u\in\mathcal{A}_0$ there exist $i\in v$ and $j\in u$ such that $v-\{i\}+\{j\}$ and $u-\{j\}+\{i\}$ are in $\mathcal{A}_0$. For example, this subclass includes multi-platform online advertising.
> - While it is true that general $\alpha$ is NP-hard to compute, in many practical applications, $\alpha$ (or any upper bound) is known due to the problem structure. For example, in online advertising with the winner's bid revealed [2] or inventory control, the feedback graph is known to have $\alpha=1$, and thus the change from $K$ to $\alpha$ is significant. Beyond practical applications, our contribution also includes a theoretical understanding of the optimal regret characterization in the combinatorial setup, including both the $\widetilde{\Theta}(S\sqrt{T}+\sqrt{\alpha ST})$ characterization on the above subclass of decision subsets and the impossibility result in Section 4.1.
> - We are grateful to the reviewer for the detailed comments and corrected typos and have carefully addressed them in the revision.
> - The references mentioned by the reviewer are indeed relevant and have been added to the revision.
> - By "tabular" contextual bandits, we refer to the contextual bandits without additional structures (e.g. linear contextual bandits) that only treat the contexts as general variables affecting the rewards.
>
> Once again, we deeply appreciate the reviewer’s insightful feedback, which has helped us refine and strengthen our work.
>
> References:
>
> [1] Lattimore, Tor and Kveton, Branislav and Li, Shuai and Szepesvari, Csaba. TopRank: A Practical Algorithm for Online Stochastic Ranking, NeurIPS 2018.
>
> [2] Han, Yanjun and Weissman, Tsachy and Zhou, Zhengyuan. Optimal no-regret learning in repeated first-price auctions. Operations Research 2025.
>
> [3] Alon, Noga and Cesa-Bianchi, Nicolo and Dekel, Ofer and Koren, Tomer. Online learning with feedback graphs: Beyond bandits. COLT 2015.

---

> > ### Comment · Reviewer_bn5t · 2025-04-05
> >
> > I would like to thank the authors for taking the time to reply to my comments and concerns.
> > While some of my remarks and minor doubts were addressed, I still have some concerns regarding some of the points I raised.
> >
> > **KL chain rule in the lower bound:**
> > Thank you for clarifying this part of the derivation in the lower bound.
> > Please, observe that you would need to correct the part of the proof at lines 165-196 (column 2) accordingly.
> >
> > **Proposed fix to the lower bound:**
> > It is yet unclear to me how the proposed fix to the lower bound would resolve the issue I raised in my previous review.
> > Constructing environments for any vector $u \\in [n]^S$ does not appear to be sufficient.
> > The only change would essentially be that the sum $\\frac{1}{n} \\sum\_{u=1}^n$ would be replaced by $\\frac{1}{n^S} \\sum\_{u \\in [n]^S}$, thus the larger variety of environments due to "uncorrelating" the index within each group of arms would well be compensated by the multiplicative $\\frac{1}{n^S}$ factor throughout the entire proof.
> > Given this, it is also not clear to me how using the stopping time argument from [1] would help in this regard.
> > Do the authors actually have a convincing argument for solving the issue in the lower bound pointed out in my review?
> >
> > **Self-loops assumption and stated claims:**
> > Thanks for clarifying the required assumption on the graph in your results; it is appreciated as I believe it will help the reader.
> > About lifting the self-loops assumption for strongly observable graphs, I would like to have more details from the authors.
> > Removing such an assumption is shown to be definitely non-trivial in previous works (e.g., [3] as also mentioned by the authors) as it typically requires specialized technical lemmas and a more refined regret analysis.
> > This is even more true in the more general combinatorial setting considered in this work.
> > One should then be careful in verifying that these aspects can indeed be achieved in a more difficult combinatorial setting, *simultaneously* to keeping the other properties of the proposed algorithm.
> > As of now, there are just not enough details in order to assess whether all these aspects can indeed be achieved and, I reiterate, it could require a non-negligible amount of work.
> >
> > **Constrained decision sets:**
> > Thank you for clarifying this aspect of your work.
> > I believe your remark on extending the applicability of your results to constrained decision sets satisfying this "exchange" property is interesting and should be included in the paper. I would also explicitly leave the interesting question of extending the results to arbitrarily constrained decision sets to future work.
> >
> > **About prior knowledge of $\\alpha$:**
> > Even if there may be some specific applications where the prior knowledge of $\\alpha$ may not be a problem, I still believe that this is a limitation of the proposed algorithm.
> > The problem setting considered here is a general combinatorial bandits setting that considers an *arbitrary* feedback graph (containing all self-loops).
> > Hence, the algorithm should be able to efficiently work in the most general case, where the prior knowledge of $\\alpha$ is not available.
> > As I already mentioned in my previous review, this assumption is *not* required in recent related work on bandits with feedback graphs and it remains a concerning computational aspect.
> >
> >
> > All things considered, it appear that solving all the issues would anyways require significant amount of work and changes to the content of the submission.
> > In general, this would often require a resubmission of the work after a major revision, hence why I am currently keeping my initial evaluation.
> > I would be happy to further discuss with the authors about my remaining doubts, especially if this may lead to resolving them, and I would also be happy to engage in a discussion with the other reviewers about my comments above.

---

> > > ### Author Response · Authors · 2025-04-05
> > >
> > > We appreciate the reviewer's in-depth response. Indeed, we make a few changes to validate and improve our results, but the only major change is the lower bound correction. Please see the changes as follows:
> > >
> > > **Strongly Observable Graphs:** When $S=1$ it reduces to MAB, and we need to invoke an adapted version of Lemma 4 in [1]. But when $S>1$ it is straightforward by noting $\sum_{i\in U}x^t_i(\hat{r}^t_i)^2 \le \sum_{i\in U}x^t_i(\frac{\sum_{j\neq i}\mathbf{1}[v^t_j=1]}{\sum_{j\neq i}x^t_j})^2 \le \sum_{i\in U}x^t_i(\frac{S}{S-1})^2 \le 4\sum_{i\in U}x^t_i \le 4S$, where $U$ is the set of nodes with no self-loop. By splitting the sum in eq.(10) into $U$ and $U^c=[K]-U$, we can apply the same upper bound to the latter. To apply Lemma D.2 on $U^c$ (nodes with self-loops), note that for $i\notin U$, $\frac{1}{\sum_{j\in N_{in}(i)}x^t_j} \le \frac{1}{\sum_{j\in N_{in}(i)\land j\notin U}x^t_j}$ and $\alpha$ of subgraphs is upper bounded by $\alpha$ of the whole graph. The final bound is changed by a constant. The intuition is that, when $S>1$, every node in $U$ has to be observed, so the problem becomes easier.
> > >
> > > **Lower Bound Correction:** Due to space limit here, we kindly point the reviewer to [2]'s Appendix D (referred as *their*) for some details. The same decomposition follows *their* eq.(6): under any environment $u$, let $\mathbb{E}^{(u)}$ denote expectation under this environment. Then
> > > $\mathbb{E}^{(u)}[\mathsf{R}(\pi)] \ge \frac{\Delta}{2}\sum_{m=1}^S T - T_m(u_m;\tau_m)$
> > > where $T_m(j;t)$ denotes the number of pulls of arm $a_{m,j}\in I_m$ up to time $t$, $T_m(t)=\sum_{a_{m,j}\in I_m}T_m(j;t)$ the total pulls in $I_m$, and $\tau_m = \min\lbrace T,\min\lbrace t: T_m(t)\ge T\rbrace\rbrace$ the stopping time as in [2]. Let $u_{-m}$ denote $u$ with $m$-th entry removed and $u^{-m}$ be its $m$-th entry replaced by $0$. As noted before, we only need to bound the Bayes regret $\mathbb{E}_{u\sim [n]^S} \mathbb{E}^{(u)}[\mathsf{R}(\pi)]$ (drawn uniformly). This is bounded by
> > >
> > > $$\frac{\Delta}{2}\sum_{m=1}^S\mathbb{E}_{u\sim [n]^S} \mathbb{E}^{(u)}[T - T_m(u_m;\tau_m)] = \frac{\Delta}{2}\sum\_{m=1}^S\mathbb{E}\_{u\_{-m}\sim [n]^{S-1}}\mathbb{E}\_{u_m\sim[n]}\mathbb{E}^{(u)}[T - T_m(u_m;\tau_m)].$$
> > >
> > > Then we use Pinsker's inequality and KL applied to the *observed* rewards (recall our reward distribution construction in Section 2; now $a_{m,u_m}\in I_m$ is optimal) similar to *their* eq.(8), and get:
> > > $$\mathbb{E}^{(u)}[T_m(u_m;\tau_m)]\le\mathbb{E}^{(u^{-m})}[T_m(u_m;\tau_m)]+4\Delta T\sqrt{\mathbb{E}^{(u^{-m})}[T_0]+\mathbb{E}^{(u^{-m})}[T_m(u_m;\tau_m)]}$$
> > > where $T_0$ denotes the pulls outside of the independent set $I=\cup_{m=1}^S I_m$. Again, if $\mathbb{E}^{(u^{-m})}[T_0]\ge \sqrt{\alpha ST}$ for any $m$, we are done, since $\mathbb{E}^{(u^{-m})}[\mathsf{R}(\pi)] \ge \frac{1}{4}\mathbb{E}^{(u^{-m})}[T_0].$ Otherwise, we have for each $m$, by Jensen's inequality,
> > > $$\mathbb{E}\_{u_m\sim[n]}\mathbb{E}^{(u)}[T -T_m(u_m;\tau_m)]\ge T - \frac{1}{n}\sum_{u_m=1}^n\mathbb{E}^{(u^{-m})}[T_m(u_m;\tau_m)]-4\Delta T\sqrt{\sqrt{\alpha ST}+\frac{1}{n}\sum_{u_m=1}^n\mathbb{E}^{(u^{-m})}[T_m(u_m;\tau_m)]}$$
> > > $$\ge T-\frac{S+T}{n}-4\Delta T\sqrt{\sqrt{\alpha ST}+\frac{S+T}{n}}.$$
> > > The second line follows from the definition of $\tau_m$ such that $\sum_{u_m=1}^nT_m(u_m;\tau_m) = T_m(\tau_m) \le T+S$ since each time the pulls increase at most by $S$. When $T\ge \max\lbrace S,\alpha^3\rbrace$ and $n\ge 4$, plugging this back, the Bayes regret is bounded by
> > > $$\frac{\Delta}{2}\sum\_{m=1}^S\mathbb{E}\_{u\_{-m}\sim [n]^{S-1}}\left[T - \frac{S+T}{n}- 4\Delta T\sqrt{\sqrt{\alpha ST}+\frac{S+T}{n}}\right]\ge\frac{\Delta ST}{4}-4\Delta^2ST\sqrt{T/n}$$
> > > which gives $\frac{3}{1024}\sqrt{\alpha ST}$ for $\Delta=\frac{1}{64}\sqrt{n/T}=\frac{1}{64}\sqrt{\alpha/(ST)}$. The key idea is that with this decoupled multi-index $u$ and the introduction of stopping time, we are allowed to deal with $S$ independent sub-problems each incurring regret of order $\sqrt{nT}$ (recall each sub-problem $I_m$ has $n$ independent arms).
> > >
> > > **Knowledge of $\alpha$:** To the best of our knowledge, we are not aware of works achieving $\tilde{O}(\sqrt{\alpha T})$ without using $\alpha$ in the parameters in the adversarial case, while it is possible for stochastic rewards (e.g. the algorithm in our Appendix B doesn't need $\alpha$). We would greatly appreciate it if the reviewer could point us to a reference, so we can include an appropriate comparison.
> > >
> > > We hope the arguments here are clear, and are more than happy to answer any questions. While we have a few changes, the major amount of change only lies in the lower bound, so we kindly ask the reviewer to reconsider the evaluation.
> > >
> > > [1] Alon, Noga and Cesa-Bianchi, Nicolo and Dekel, Ofer and Koren, Tomer. Online learning with feedback graphs: Beyond bandits. COLT 2015.
> > >
> > > [2] Lattimore, Tor and Kveton, Branislav and Li, Shuai and Szepesvari, Csaba. TopRank: A Practical Algorithm for Online Stochastic Ranking, NeurIPS 2018.

---

### Official Review · Reviewer_nVDS · 2025-03-17

**Overall Recommendation:** 4

**Summary:**

The paper studies the problem of semi-bandits, aiming to generalize bandit feedback and the full-information setup into a single framework. The problem is formulated as follows: Given a graph $G=(K,E)$, where $K$ represents the number of vertices and
$E$ is the set of edges, consider a $K$-armed bandit problem. If the learner selects an arm
$a\in[K]$], they receive all the rewards associated with the neighbors of arm $a$ in $G$. When $G$ is a complete graph, this setting corresponds to the full-information setup, and when $G$ only contains self-loop edges, the feedback procedure corresponds to a standard multi-armed bandit problem.

In the full-information setup and multi-armed bandits (the two extreme cases), the problem is well understood. The main objective of this paper is to bridge the gap between these two cases by providing upper and lower bounds that characterize the minimax regret based on the properties of $G$. The paper establishes a lower bound and, under the assumption that there exists a probability density function over the decision set with certain correlation properties (outlined in Theorem 3.2), proposes a mirror descent-type algorithm that achieves a matching upper bound. Furthermore, the authors discuss the necessity of the correlation property of the probability density function in the final section of the paper.

**Claims And Evidence:**

All claims in the paper have been completely addressed.

**Essential References Not Discussed:**

I do not have in mind any essential references that the authors did not discuss.

**Experimental Designs Or Analyses:**

The paper is theoretical, so no experimental design has been considered.

**Methods And Evaluation Criteria:**

The method and evaluation criteria make sense.

**Other Comments Or Suggestions:**

1. I believe the authors should elaborate more on some definitions that were mentioned in the paper. This would enhance the readability of the paper: $S$ on the first page, defined as the size of the combinatorial decisions.

It would also be helpful if the authors provided the definition of independent and dominating subsets in Section 1.3.

**Other Strengths And Weaknesses:**

The paper appears theoretically solid. I have some comments regarding the presentation, which I will address in the next section.

**Questions For Authors:**

1. The algorithm requires knowledge of $\alpha,$ as it provides the sufficient information the learner needs about the graph
$G$. Do the authors have any comments or insights on adaptive methods where the learner has no prior information about
$G$?

2. Do the authors have any comments on problem-dependent bounds, such as a first-order regret bound?

3. Have the authors explored other regularizers for the mirror map, such as Tsallis entropy or the log-barrier function? Is there any advantage to using negative entropy, as stated in the current algorithm?

**Relation To Broader Scientific Literature:**

The contribution of the paper is related to the bandit framework and online machine learning setup, which has been addressed by the authors.

**Theoretical Claims:**

I have checked the correctness of the proofs in the main body of the paper. The theoretical claims are well-supported, and the methodology is sound.

---

> ### Author Rebuttal · Authors · 2025-03-30
>
> We truly appreciate the reviewer's review and inspiring questions. Please see the following for our updates in light of the review and our thoughts:
> - We appreciate the review's point on the readability and in our revision, we have added a few sentences on $S$ and the fact that $S>1$ corresponds to enlarging the selection budget of the learner in the classic multi-armed bandit case. We also add the definition of independent and dominating subsets in Section 1.3.
>
>
> - Q1 is an important question and requires answers from 2 aspects:
> 1. Suppose the graph $G$ is given. It is known to be NP-hard to find $\alpha$ for any $G$. However, one can substitute any upper bound of $\alpha$ into the algorithm and the results remain valid. More importantly, in many applications, $\alpha$ is (perfectly or approximately) known due to the problem structure. For example, in online advertising with the winner's bid revealed [1] or inventory control, the feedback graph is known to have $\alpha=1$, and thus the change from $K$ to $\alpha$ is significant.
> 2. If $G$ is not known but fixed. It takes the learner $K$ rounds to pull every node to know $G$ and pay $K$ regret, so our results still hold. Suppose $G$ is not known and is time-varying and $G_1,...,G_T$ are chosen adversarially. When the rewards are adversarial as in our case, [2] shows a lower bound $\Omega(\sqrt{KT})$ for $S=1$, i.e. no graph information can be leveraged, so we do not expect an improvement in our $S>1$ case either. When the rewards are stochastic, [2] shows a bound $\widetilde{O}(\sqrt{\alpha T})$, so it is interesting to see if similar improvement is possible for the combinatorial case with stochastic rewards.
>
> - Q2: To the best of our knowledge, first-order bounds in the presence of feedback graphs are still unexplored even in the multi-armed bandit setting ($S=1$), and we do not have much to say in this even harder combinatorial setting for now. Meanwhile, under stochastic rewards, instance-dependent bounds are possible, and how to leverage $G$ there would be an interesting future direction.
>
> - Q3: We choose to work with negative entropy only because it is relatively common. It is yet known that (related to Q2) log-barrier leads to first-order bound in the MAB setting. Thus there may be practical advantages or theoretical advantages if one is interested in the first-order bound.
>
> We thank the reviewer for the review and welcome any follow-up questions or suggestions.
>
> References:
>
> [1] Han, Yanjun and Weissman, Tsachy and Zhou, Zhengyuan. Optimal no-regret learning in repeated first-price auctions. Operations Research 2025.
>
> [2] Cohen, Alon and Hazan, Tamir and Koren, Tomer. Online learning with feedback graphs without the graphs. ICML 2016.

---

### Decision · Program_Chairs · 2025-05-01

**Decision:**

Accept (poster)

**Comment:**

The paper studies adversarial combinatorial semi-bandits with feedback graphs, with the aim to interpolate regret bounds between the standard full-information and semi-bandit settings (similarly to what has been done for non-contextual bandits).

Some errors were found during the reviewing process, but during the discussion period, the authors gave enough information to convince the reviewing team that the arguments could be fixed. With this issue resolved, I recommend acceptance under the constraints that all points brought up during the discussion period (including but not limited to the correctness of the claims) are addressed in the revised version.